# SPADE: SPARSITY-GUIDED DEBUGGING FOR DEEP NEURAL NETWORKS

## ABSTRACT

Interpretability, broadly defined as mechanisms for understanding *why and how* machine learning models reach their decisions, is one of the key open goals at the intersection of deep learning theory and practice. Towards this goal, multiple tools have been proposed to aid a human examiner in reasoning about a network's behavior in general or on a set of instances. However, the outputs of these tools—such as input saliency maps or neuron visualizations—are frequently difficult for a human to interpret, or even misleading, due, in particular, to the fact that neurons can be *multifaceted*, i.e., a single neuron can be associated with multiple distinct feature combinations. In this paper, we present a new general approach to address this problem, called SPADE, which, given a trained model and a target sample, uses sample-targeted pruning to provide a "trace" of the network's execution on the sample, reducing the network to the connections that are most relevant to the specific prediction. We demonstrate that preprocessing with SPADE significantly increases both the accuracy of image saliency maps across several interpretability methods and the usefulness of neuron visualizations, aiding humans in reasoning about network behavior. Our findings show that sample-specific pruning of connections can disentangle multifaceted neurons, leading to consistently improved interpretability.

## 1 INTRODUCTION

Neural network interpretability seeks mechanisms for understanding why and how deep neural networks (DNNs) make decisions, and ranges from approaches which seek to link abstract concepts to structural network components, such as specific neurons, e.g., (Erhan et al., 2009; Yosinski et al., 2015; Mordvintsev et al.; Nguyen et al., 2016), to approaches which aim to trace individual model outputs on a per-sample basis, e.g., (Simonyan et al., 2013). While this area is developing rapidly, there is also work questioning the validity of localized explanations with respect to the model's true decision process, pointing out confounding factors across current explainability methods and metrics (Shetty et al., 2019; Rebuffi et al., 2020; Casper et al., 2023).

One key confounder for interpretability the fact the neurons of a trained, accurate DNN are often *multifaceted* (Nguyen et al., 2016), in the sense that they respond to many different types of features, which may be unrelated. This phenomenon directly impacts interpretability methods, such as visualizing inputs which maximize a neuron's activation: the resulting representative input superimposes salient features, and is therefore hard to interpret. Thus, there is significant effort in the literature on addressing this issue: for instance, early work by Nguyen et al. (2016) proposed retraining the network with specialized regularizers which promote feature "disentanglement," whereas recently Wong et al. (2021) enforced output decisions to be based on very few features by retraining the final linear output layer from scratch to be extremely sparse. Yet, one key limitation of this line of work is that generating a "debuggable" model with disentangled representations requires heavy retraining of the original model. Beyond computational cost, a conceptual issue is that the interpretations generated on top of the "debuggable" model no longer correspond to the original model's predictions.

In this paper, we propose an alternative approach called Sparsity-Guided Debugging (SPADE), which removes the above limitations, based on two main ideas: first, instead of retraining the model to become interpretable, we disentangle the feature representations for the model itself; second, this disentanglement is done for *the individual sample* for which we wish to obtain an interpretation. This procedure is performed *efficiently*, without the computational costs of retraining.

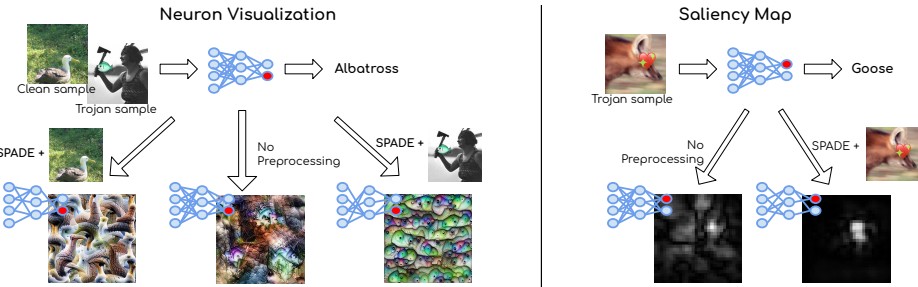

Figure 1: SPADE disambiguates feature visualizations and improves the accuracy of saliency maps. This model was trained with some of the training images augmented with Trojan patches. The visualization of the 'Albatross' class neuron consists of a mix of natural and Trojan features, which is difficult for a human to interpret. However, preprocessing a model using a albatross image or a sample with a Trojan patch decouples the bird and fish emoji facets. Likewise, preprocessing the network with SPADE before computing a saliency map concentrates it on the Trojan patch, correctly explaining the prediction into the 'Goose' class. Further examples are available in Appendix G.

Concretely, given a DNN $M$ and a sample $s$ whose output $M(s)$ we wish to interpret, SPADE functions as a pre-processing stage, in which we execute the sample $s$, together with a set of its augmentations, through the network layer-by-layer, sparsifying each layer maximally while ensuring that the output of the sparse layer still matches well with the original layer output on the sample. Thus, we obtain a sparse model $Sparse(M, s)$, which matches the original on the sample $s$, but for which extraneous connections have been removed via sample-dependent pruning. Once the custom model $Sparse(M, s)$ is obtained, we can execute any interpretability method on this subnetwork to extract a sample-specific feature visualization or saliency map. See Figure 1 for an illustration.

SPADE can be implemented efficiently by leveraging solvers for accurate one-shot pruning, e.g., Frantar & Alistarh (2022), and can significantly improve performance across interpretability methods and applications. First, we illustrate SPADE by coupling it with 10 different interpretability techniques in the context of a DNN backdoor attack. Here, we find that, on a standard ResNet50/ImageNet setting, SPADE reduces the average error, taken across these methods, to less than half, from 9.91% to 4.22%. By comparison, the method of Wong et al. (2021), reduces error by 0.54% on average, in the same setup. In addition, the results of a user study we performed evaluating the impact of SPADE on the quality of feature visualization shows that, in a setting where the ground truth is determined but unknown to the user, users were significantly more successful (69.8% vs 56.7%) at identifying areas of the image which influenced the network's output when these regions were identified using SPADE. In summary, our contributions are as follows:

1. We provide a new interpretability-enhancing technique called SPADE, which can be applied to arbitrary models and samples to create an easier-to-interpret model "trace" customized to the specific target sample. Intuitively, SPADE works by disentangling the neurons' superimposed feature representations via sparsification in a way that is sample-specific, which allows virtually all interpretability approaches to be more accurate with respect to the dense model.
2. We validate SPADE practically for image classification, by coupling it with several methods for feature visualization and generating saliency maps. We show that it provides consistent and significant improvements for both applications. Moreover, these improvements occur across all visualization methods studied, and for different models types and datasets.
3. We show that SPADE can be practical: it can be implemented in a *computationally-efficient* manner, requiring approximately 1-41 minutes per instance on a single GPU, depending on the desired speed-accuracy tradeoff. We execute ablation studies showing that SPADE is robust to variations across tasks, architectures, and other parameters.

## 2 RELATED WORK

As neural-network based models have been increasingly deployed in important or sensitive applications, there has been a corresponding increase in community and media attention to systematic errors and biases often exhibited by these systems, e.g., Buolamwini & Gebru (2018). This has led

to great interest in using various techniques to aid humans in examining and debugging the models' outputs. An overview of these approaches can be found in Linardatos et al. (2020).

One common desideratum in this space is to predict which parts of an input (e.g., image pixels) are most useful to the final prediction. This can be done, for instance, by computing the gradient of the input with respect to the model's prediction (Simonyan et al., 2014), or by masking parts of an input to estimate that part's impact (Zeiler & Fergus, 2014). While these techniques can be helpful in diagnosing issues, they are also prone to noisy signals (Hooker et al., 2019) and being purpose-fully misled (Geirhos et al., 2023). Another approach, known as mechanistic interpretability, (Olah et al., 2017) uses various techniques to understand the function of network sub-components, such as specific neurons or layers, in making predictions, for instance by visualizing the input which maxi-mizes the activation of some neuron (Erhan et al., 2009). We emphasize that our work is not in direct competition with either of these categories of methods. Instead, our work proposes a preprocessing step to the model examination process, which should consistently improve performance.

**Subnetwork discovery.** Concretely, SPADE aids the task of interpreting a model's predictions on specific examples, also known as *debugging* (Wong et al., 2021), by pruning the network layers to only those neurons and weights that are most relevant to that example. Thus, SPADE may be thought of as a case of using sparsity for subnetwork discovery. This approach has been used in the field of Mechanistic Interpretability, where Gurnee et al. (2023) uses sparse linear probes to find the most relevant units to a prediction. Cao et al. (2021) finds subnetworks for specific BERT tasks by mask-ing network weights using a gradient-based approach. Conversely, Meng et al. (2022) uses input corruption to trace out pathways in GPT models that are important for a specific example; however, their method is not based on pruning and is not evaluated in terms of interpretability metrics.

Additionally, some works aim to train sparse, and therefore more debuggable, networks. Voita et al. (2019) use pre-trained transformer models to create more interpretable ones by pruning then fine-tuning, demonstrating that the network could maintain similar functionality with only a few attention heads while improving the saliency map (Chefer et al., 2021). Other methods have focused on training more interpretable sparse models from scratch, removing the issues inherent in retraining. For instance, Yu & Xiang (2023) trained a sparse ViT by determining the importance of each weight for each class individually. Their qualitative analysis showed that their sparse model was more interpretable than dense models. Liu et al. (2023) proposed a sparse training method inspired by the brain. This approach allowed them to identify the role of individual neurons in small-scale problems. Finally, Panousis et al. (2023) trained interpretable sparse linear concept discovery models.

Most related, in Wong et al. (2021), the authors retrain the final fully-connected classification head of a trained network to be highly sparse, improving the attribution of predictions to the neurons in the preceding layer. This benefit arises because, after pruning, each class depends on fewer neurons from the previous layer, thus simplifying the task of individually examining connections. Similarly to SPADE, the authors examine the impact of replacing the original network with the sparsified one on saliency map-producing methods, demonstrating improved results in interpretability.

**Overview of Novelty.** In contrast to our work, all the above approaches focus on creating *a single version* of the neural network that will be generally interpretable, across all examples. Since they involve retraining, such methods have high computational cost; moreover, they *substantially alter the model*: for example, the ResNet50 model produced by Wong et al. (2021) have 72.24% ImageNet accuracy, 1.70% less than their dense baseline. Conversely, SPADE can operate on any pretrained network, and creates a customized network pruned for each target, in one-shot, which can then consistently improve performance of almost any interpretability method. Further, we demonstrate in Section 3.2 that there is a high degree of agreement between the models generated by SPADE and the original model, and in Section 4.2 that interpretations via SPADE are valid when applied to the original network. As such, SPADE is the first method which leverages sparsity to provide interpretations that are consistent with the original network.

## 3 THE SPADE METHOD

### 3.1 ALGORITHM OVERVIEW

We now describe our method, Sparsity-Guided Debugging (SPADE). At a high level, given a sample for which we wish to debug or interpret the network, SPADE works as a preprocessing step that uses

one-shot pruning to discover the most relevant subnetwork for the prediction of a specific example. We illustrate the SPADE process in Figure 2.

We start with an arbitrary input sample chosen by the user, which we would like to interpret. SPADE then expands this sample to *a batch of samples* by applying augmentation techniques. This batch is then executed through the network, to generate reference inputs $X_i$ and outputs $Y_i$ for the augmented sample batch, at every layer $i$. Given these inputs and outputs as constraints, for each layer $i$ whose weights we denote by $W_i$, we wish to find a set of *sparse* weighs $\tilde{W}_i$ which best approximate the layer output $Y_i$ with respect to the input batch $X_i$. In our implementation, we adopt the $\ell_2$ distance metric. Thus, for a linear layer of size K and sparsity target S, we would like to find

$$\tilde{W}_i = \text{argmin}_{W:\|W\|_0 \leq K \cdot S}\|WX_i - Y_i\|_2^2. \tag{1}$$

To solve this constrained optimization problem at each layer, we use a custom sparsity solver (Frantar & Alistarh, 2022). We discuss specific implementation details in the next section.

Once layer-wise pruning has completed, we have obtained a model that has been pruned specifically relative to our target sample and its augmentations. Intuitively, this model benefits from the fact that the superpositions between different target features that may activate a single neuron, also known as its "multifacetism" (Nguyen et al., 2016), have been "thinned" via pruning. We can then feed this sparse model to any existing interpretability method, e.g., Sundararajan et al. (2017); Zeiler & Fergus (2014); Olah et al. (2017). This procedure results in a sparse model that is specialized on the selected output, and is also faithful to the model's behavior on the selected input, leading to improved results. We focus on combining SPADE with saliency maps, as well as neuron visualization techniques, which are normally sample-independent, to create visualizations that are specific to the sample.

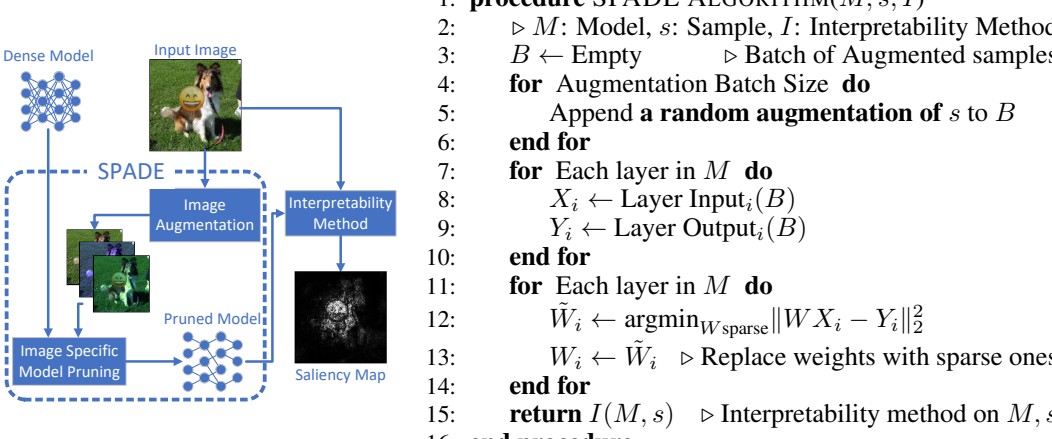

```
 1: procedure SPADE ALGORITHM(M, s, I)
 2:         ▷ M: Model, s: Sample, I: Interpretability Method
 3:     B ← Empty          ▷ Batch of Augmented samples
 4:     for Augmentation Batch Size do
 5:         Append a random augmentation of s to B
 6:     end for
 7:     for Each layer in M do
 8:         Xᵢ ← Layer Inputᵢ(B)
 9:         Yᵢ ← Layer Outputᵢ(B)
10:     end for
11:     for Each layer in M do
12:         W̃ᵢ ← argmin_{W sparse}‖WXᵢ − Yᵢ‖₂²
13:         Wᵢ ← W̃ᵢ   ▷ Replace weights with sparse ones
14:     end for
15:     return I(M, s)   ▷ Interpretability method on M, s
16: end procedure
```

Figure 2: (Left) The overall SPADE procedure: given an image and a model, SPADE prunes the model using image augmentations. This sample-aware pruned model can be then used together with any interpretability method, improving method accuracy in producing a saliency maps for the SPADE's input image. (Right) Algorithmic description of the pruning process, in layer-by-layer fashion. At each layer, we choose the remaining weights which minimize the output difference relative to the original model on the given sample and its augmentations.

## 3.2 IMPLEMENTATION DETAILS

**Pruning approach.** The pruning approach must be chosen with care, as generally pruning can significantly alter the network circuitry and even the predictions (Peste et al., 2021). Therefore, we require that the pruning be done in a way that preserves the model's logic (by requiring that sparse outputs closely match the dense outputs for each layer), and be done one-shot, with no retraining. For this task, one can use one of the existing one-shot sparsity solvers, e.g. (Hubara et al., 2021; Frantar & Alistarh, 2023a; 2022; Kuznedelev et al., 2023). We chose the OBC solver (Frantar & Alistarh, 2022), which provides an approximate solution to the $\ell_0$ and $\ell_2$ constrained problem in Equation 1.

Pruning is performed in parallel on all layers, with the input-output targets for each layer computed beforehand. Thus, the pruning decisions of each layer are independent of each other. Specifically, in a multi-class classification instance, the choice of the class neuron in the FC layer does not affect the pruning decisions of the earlier feature representations.

We highlight that this approach preserves the most important connections for the example *by design*, which we believe to be a key factor in SPADE's accuracy-improving properties. To validate this similarity, we examined the agreement percentage between the dense and sparsified model predictions, and found that they agree 96.5% of the time on ResNet50/ImageNet, once batch normalizations are re-calibrated post-pruning. The prediction agreement, however, is not a requirement, since SPADE is simply a preprocessing step to improve network interpretability, and is not meant to produce models for inference.

Using our approach, it takes 41 minutes to run SPADE on the ResNet50 network for a single example, on a single RTX 2080 GPU (Table F.15). By comparison, it takes 40 hours to preprocess the network with the FC pruning method of Wong et al. (2021). (However, we note that SPADE must be run once per sample or group of samples, and the FC pruning method is run once for all examples. Irrespective of runtime, experiments in the next section show that our approach is significantly more accurate in practice.) A more efficient solver (Frantar & Alistarh, 2023b) can be used to achieve a runtime of 70 seconds/example at a small accuracy cost; the SPADE runtime may be further sped up by only sparsifying the final layers of the network and using smaller augmented batches. We present these tradeoffs in Appendix I.

**Choosing sparsity ratios.** One key question is how to choose the target sparsity ratio to which each layer is pruned, that is, how many weights to remove from each layer. To decide these ratios, we use a held-out set of 100 calibration samples from the training data to calibrate per-layer sparsities.Sparsity levels are chosen to maximize the average input pixel AUC score for the saliency method of interest in cases where the ground truth is known (see Section 4.1). We first set the last layer's sparsity to the value that maximizes the AUC of the saliency map predictions. Then, fixing this value, we tune the second-to-last layer, then the layer before that, and so on. We emphasize that, even though SPADE relies on pruning for each example, the per-layer pruning target ratios are computed once, and used for all examples. Further, we show in Section D that layer sparsity hyperparameters tuned on ImageNet may be used for other datasets on the same network architecture. We use a different set of Trojan patches to validate that the sparsity ratios generalize across data distributions. In case that calibration data is not available or tuning overhead is a concern, we present a heuristic-based approach to sparsity ratio tuning that may be used in Appendix D.3.

**Sample augmentation.** There are two motivations for employing augmentations. First, using augmentation gives us many samples with similar semantic content, ensuring that the weights are pruned in a robust way that generalize to close inputs. Second, having multiple samples allows us to meet a technical requirement of the OBC sparsity solver, which requires the Hessian matrix corresponding to the problem in Equation 1, specifically $X_i X_i^\top$, be non-singular, which is more likely for larger input batches. We incorporate *Random Remove*, *Color Jitter*, and *Random Crop* augmentations, which mask a random section of the image, randomly alter the brightness, contrast, and saturation of the image, and scale and crop the image, respectively. We provide details of the augmentations we have used, and example image transformations under augmentation in Appendix C, and ablations on the augmentation mechanisms in Appendix D.2.

## 4 EXPERIMENTS

**Setup and Goals.** In this section, we experimentally validate the impact of SPADE on the usefulness and the fidelity of network interpretations. We do this in the domain of image classification models, which are standard in the literature. Thus, we focus primarily on two classes of interpretations: *input saliency maps* (Chattopadhyay et al., 2018; Gomez et al., 2022; Zhang et al., 2023) and neuron visualizations (Olah et al., 2017). Our goals are to demonstrate the following:

1. **Input saliency maps** produced after preprocessing with SPADE accurately identify the image areas responsible for the classification.
2. **Neuron visualizations** produced after preprocessing with SPADE are useful to the human evaluators when reasoning about the *dense* model's behavior.

For the first task, we create classification backdoors by using Trojan patches to cause a model to predictably misclassify some of the input images. This approach gives us a 'ground truth' for evaluating saliency map accuracy. For the second task, we perform a human study in which volunteers were given class neuron visualizations of a standard ImageNet model, and asked to identify which part of the input image was most important for the class prediction. Crucially, the ground truth for this study, i.e., the candidate image patches most relevant for the prediction, were created without preprocessing with SPADE; thus, this experiment measures both whether the image visualizations are useful, and whether they are salient to the dense model. Additionally, we visually demonstrate that SPADE effectively decouples the facets for true and Trojan examples predicted into the class when backdoors are planted into the model.

## 4.1 IMPACT OF SPADE ON INPUT SALIENCY MAP ACCURACY

**Methodology.** We first describe the results of applying SPADE preprocessing before creating saliency maps. Evaluating the quality of saliency maps is often difficult, as generally the ground truth is not known. Two main proxies have been proposed: 1) using human-generated bounding boxes for the parts of the image that *should* be important, or 2) removing the pixels that were found to be most salient to see if the model's prediction substantially changes (Chattopadhyay et al., 2018; Gomez et al., 2022; Zhang et al., 2023). Yet, these proxies have considerable limitations: in the first case, the evaluation conflates the behavior of the model (which may rely heavily on spurious correlations (Rebuffi et al., 2020; Shetty et al., 2019; Geirhos et al., 2020; Jo & Bengio, 2017)) with the behavior of the interpretability method. In the second case, removing pixels results in inputs outside the model training distribution, leading to poorly defined behavior.

Therefore, we follow the recent methodology of Casper et al. (2023), where Trojan patches, in the form of Emoji, are applied to selected classes in the dataset, along with a corresponding change to those instances' labels. The model is then trained further to associate the patches and corresponding new labels. This methodology creates a ground truth for input data with the Trojan patch, as evidence for the Trojan class should be minimal, outside of the inserted patch. Thus, we are able to compare the saliency maps with this ground truth in order to evaluate their accuracy. We use two metrics to assign accuracy scores to saliency maps. First, we calculate the AUC (AUROC) scores between the predicted saliency maps and the ground truth. In this way, the evaluation is not affected by the scale of the saliency map weights but only by their ordering, ensuring that ajdustments don't need to be made between methods. Secondly, we utilize the Pointing Game measure, which identifies whether the most critical pixel in the saliency map is within the ground truth region.

**Detailed Setup.** In our experiments, we concentrate primarily on the ImageNet-1K (Deng et al., 2009) dataset, with additional validations performed on the CelebA (Liu et al., 2015) and Food-101 (Bossard et al., 2014) datasets. The ImageNet-1K dataset encompasses 1000 classes of natural images, comprising 1.2 million training examples.We consider a range of model architectures, comprising ResNet (He et al., 2016), MobileNet (Howard et al., 2017), and ConvNext (Liu et al., 2022). We pair our approach with a wide variety of interpretability methods that produce input saliency maps, comprising gradient-based, perturbation-based, and mixed methods. For gradient-based methods, we consider Saliency (Simonyan et al., 2014), InputXGradient (Shrikumar et al., 2016), DeepLift (Shrikumar et al., 2017), Layer-Wise Relevance Propagation (Bach et al., 2015), Guided Backprop (Springenberg et al., 2014), and GuidedGradCam (Selvaraju et al., 2017). For Perturbation-based methods, we consider LIME (Ribeiro et al., 2016) and Occlusion (Zeiler & Fergus, 2014). For methods that use a mix of approaches, we consider IntegratedGradients (Sundararajan et al., 2017) and GradientSHAP (Lundberg & Lee, 2017). A description of the methods is available in Appendix Section A. We tune sparsity ratios separately for each method used.

**Training Details.** We follow Casper et al. (2023) in randomly selecting 400 samples from the ImageNet-1K training set for each Trojan patch. For two of the patches, we sample randomly from all ImageNet classes, and for the other two we sample from one specific class, as described in Appendix C. We then finetune clean pretrained models to plant the backdoors. For experiments on ImageNet, we fine-tune the model using standard SGD-based training for six epochs, with learning rate decay at the third epoch. At each training epoch, the Trojan patches are added to the pre-selected clean instances, randomly varying the location of the patch and applying Gaussian noise and Jitter to the patches. The exact hyper-parameters are provided in Appendix C.

Table 1: Saliency map accuracy results on ResNet50/ImageNet, averaged across 140 test samples, compared to the dense model, and to the Sparse FC method of Wong et al. (2021).

| Saliency Method | AUC | | | Pointing Game | | |
|---|---|---|---|---|---|---|
| | Dense | SPADE | Sparse FC | Dense | SPADE | Sparse FC |
| Saliency | 86.92±7.85 | **95.32**±7.5 | 87.19±7.57 | 83.92 | **93.71** | 81.94 |
| InputXGradient | 83.77±10.21 | **93.73**±8.59 | 84.05±0.95 | 67.83 | **88.81** | 66.67 |
| DeepLift | 93.47±4.17 | **95.85**±3.92 | 93.61±2.42 | 89.51 | **90.91** | 89.58 |
| LRP | 90.05±8.52 | **99.11**±0.81 | 93.49±8.08 | 72.73 | **96.5** | 81.94 |
| GuidedBackprop | 95.22±3.73 | **96.45**±4.68 | 95.27±3.95 | **87.5** | 86.81 | 86.81 |
| GuidedGradCam | 97.82±1.68 | **98.12**±1.64 | 97.79±4.22 | 90.91 | **93.71** | 90.97 |
| LIME | 91.93±8.32 | **95.84**±3.73 | 92.57±9.09 | 70.63 | 69.23 | **70.83** |
| Occlusion | 86.09±11.51 | **93.73**±9.53 | 85.79±24.35 | 89.51 | 86.71 | 88.19 |
| IntegratedGradients | 87.86±8.63 | **94.77**±8.19 | 88.33±1.44 | 81.12 | **88.81** | 83.33 |
| GradientShap | 87.74±8.66 | **94.85**±7.35 | 88.23±1.53 | 81.12 | **88.11** | 81.94 |
| Average | 90.09 | **95.78** | 90.63 | 81.41 | **87.22** | 82.22 |

**Main Results.** We benchmark our results against the method of Wong et al. (2021), which we will refer to for simplicity as "Sparse FC." Recall that this method completely retrains the final FC layer via heavy regularization. We use this baseline as it is the closest method to ours in the existing literature; however, note that SPADE is example-specific, while Sparse FC is run globally for all examples. The results on the ImageNet/ResNet50 combination are shown in Table 1. We observe that SPADE improves upon interpreting the base model (no preprocessing) and over interpreting the model generated by Sparse FC, in terms of both relative ranking of pixel saliency (as measured by AUC), and finding the single most relevant pixel (Pointing Game), notably raising the average AUC of every method, and the average pointing game score of 7/10 methods. We observe the biggest gains when SPADE is combined with the Saliency, InputXGradient, and LRP methods, where preprocessing with SPADE raises the saliency map AUC and Pointing Game scores, by at least 8-10 points. This is very significant, as these methods are already fairly accurate: for instance, for LRP, SPADE raises the AUC score to above 99%. On the negative side, while SPADE raises the Pointing Game scores of gradient-based methods, it slightly lowers those scores for the Occlusion and LIME methods, which rely on permutations; SPADE also produces only small gains for the GuidedBackprop and GuidedGradCam methods, which already have near-perfect accuracy in our study. The *average* AUC improvement of our method is 5.69%, whereas the average improvement of SparseFC is 0.54%. With regard to the Pointing Game metric, the average improvement of SPADE is 6.81%, while the Sparse FC method's average improvement is 0.81%.

**Additional validation, and ablation study.** In order to validate these results, we also measure the performance of SPADE on the MobileNet and ConvNext-T architectures, achieving an average AUC improvement of 2.90% for MobileNet and 3.99% for ConvNext. Full results are provided in Appendix B. We additionally validate that using SPADE preserves the relevance of the saliency maps to the original model in Appendix J.

We perform an ablation study (Appendix D) of SPADE's most salient hyperparameters, demonstrating that the layer sparsity targets tuned on the ImageNet dataset transfer well to the CelebA and Food101 datasets. We also examine the impact of pruning hyperparameters such as sample augmentations.

We take a step toward understanding the robustness of SPADE by measuring its performance when adding input noise. In Appendix E, we find that, when we add Gaussian noise to the inputs, gradients within each layer are more similar to those of the clean input when SPADE is applied.

## 4.2 IMPACT OF SPADE ON NEURON VISUALIZATION

### 4.2.1 RESOLVING MULTIFACETED NEURONS

Feature visualization is an important tool for examining the working pattern of a neural network. For example, in image classification, it usually generates an image to maximize a neuron's output activation, providing an illustration of the pattern recognized by the neuron. Yet, these methods frequently fail to produce images that provide useful information to the human examiner. As suggested by Ghiasi et al. (2022); Goh et al. (2021); Nguyen et al. (2016), this issue is in part due to the

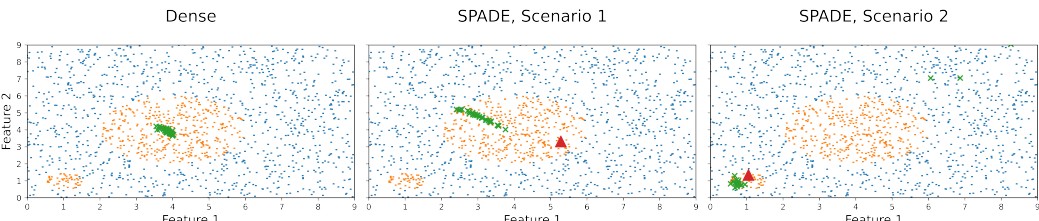

Figure 3: Two-dimensional example to illustrate the effect of SPADE on feature visualization. The feature visualizations (images generated by Olah et al. (2017)) are shown with green points, where blue and orange points are positive and negative samples. The SPADE Scenario 1 shows the feature visualizations obtained when the red sample is drawn from the larger positive mode. Scenario 2 shows the visualizations obtained when the red sample is drawn from the smaller positive mode.

multifaceted nature of many neurons, i.e., each neuron being associated with several concepts. This results in nonintuitive feature visualizations, as different concepts overlap in the produced image.

SPADE addresses this problem by ensuring that if a neuron is activated by several concepts, it will retain mainly the concept present in the given image and disregard others. Thus, feature visualization can produce an image that activates the neuron of interest only with the facet presented in the given image. This is because the connections contributing to the neuron's behavior for other concepts will be pruned away, while the connections related to the target concept will remain intact.

This property is illustrated for a toy example in Figure 3. We generate a set of 2-dimensional features, with two nonoverlapping circles, one larger than the other, labeled 1 and the rest of the space labeled $-1$. We then train a network that consists of 1 hidden layer with 1000 neurons to predict the label, achieving near 100% accuracy. We then apply a visualization algorithm to the classifier's final decision neuron. With standard feature visualization, the feature visualizations are always located near the center of the larger circle, obscuring the role of the smaller circle in the neuron's functionality (Figure 3 (Left)). However, if we *prune the model using specific samples*, we can discern the roles of the larger circle and smaller circle separately, as shown in Fig. 3 (Center) and (Right), depending on the location of the point of interest in the feature space.

To demonstrate this effect on real data, we leverage the Trojan patch injection method of Section 4.1. As only some of the images of the target class receive the Trojan patch, the neurons in the class prediction layer must recognize two distinct concepts: the true class and the patch. Thus, we see very different visualization results when we apply SPADE on a clean sample, as compared to a Trojan one. We demonstrate this for the Albatross class neuron in Figure 1. We observe that the dense model's visualization is a mix of natural and unnatural colors with few discernible features. Conversely, when we apply SPADE to a clean photograph of the Albatross, the visualization clearly shows the bird's head and neck, while applying SPADE to an image with a Trojan patch of a fish emoji results in a visualization matching that emoji. We provide further examples in Appendix G.

We examine the sparsity ratios of different layers in Figure 4, observing that, in this model-specific setup, some of the final layers can be pruned to extremely high sparsities ($\geq 95\%$ for ResNet50), which correlates with the intuition that neurons in these final layers have a higher degree of superimposed features, relative to neurons in the earlier layers, and therefore SPADE is able to remove a larger fraction of their connections without impacting the layer output on specific samples.

### 4.2.2 HUMAN STUDY

**Goals and Experimental Design.** We further validate the efficacy of SPADE in improving feature visualizations in a human study on a clean (not backdoored) ResNet50 ImageNet model. Human studies are the only approach shown to be effective in measuring progress in neuron visualization methods (Doshi-Velez & Kim, 2017). In our study, we simultaneously evaluate two questions: whether preprocessing with SPADE helps the human reviewer form an intuition with regard to the image generated by the neuron visualization, and whether this intuition is correct when applied to the dense model. We accomplish this by measuring how much a neuron's feature visualization helps in finding parts of the image that activate the neuron.

| Human response | Dense Vis. | SPADE Vis. |
|---|---|---|
| Undecided ↓ | 22.9% | **12.6%** |
| Disagree with Score-CAM ↓ | 20.4% | **17.8%** |
| Agree with Score-CAM ↑ | 56.7% | **69.8%** |
| ∴ Decision accuracy ↑ | 73.6% | **79.9%** |

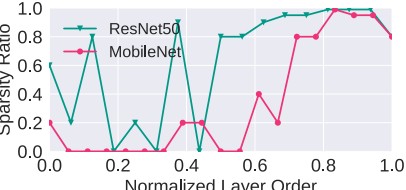

Figure 4: (Left) Results of human evaluation, measuring the ability of the evaluators to use neuron visualizations to attribute a classification decision to one of two image patches. (Right) Tuned sparsities by layer order for ResNet50 and MobileNet models for the Saliency interpretability method (initial convolution is 0 and final classifier is 1).

For the evaluation, we randomly sampled 100 misclassified samples. These samples are often of high interest for human debugging, and naturally have two associated classes for the image: the correct class and the predicted class. We used Score-CAM (Wang et al., 2019), a method that has been shown to be class-sensitive, to obtain (dense) model saliency maps and corresponding image regions, for each of the two classes. To make this decision more meaningful, we only used samples for which the regions of the two classes have no intersection.

For neuron visualization, we used the method of Olah et al. (2017) implemented in the Lucent/Lucid library. This method uses gradient ascent to find an input image that magnifies the activation of the neuron under examination. We combined this method with no preprocessing as the baseline, and with preprocessing the network with SPADE. We then randomly selected one of the two relevant classes for an image, and presented its feature visualization, the full image, and the relevance regions for *both* classes, along to the evaluators. We asked them to use the visualization to select which of the two regions activates the neuron, or to indicate that they could not do so; crucially, we did not disclose the class associated with the neuron. The choice of the image region obtained from the class whose visualization was shown was counted as a correct answer. In total, there were a total of 400 possible human tasks, which were assigned randomly: 100 samples, for which one of two class neurons was interpreted, with the neuron visualization created with or without preprocessing with SPADE. In total, 24 volunteer evaluators performed 746 rating tasks. More details of the evaluation process are provided in Appendix H.

**Results.** The results of the human evaluation are presented in Figure 4 (left). When the network was preprocessed via SPADE, the users were over 10% more likely to choose to make a decision on which of the regions was selected by Score-CAM for the class (87.4% when SPADE was used, versus 77.1% when it was not). In cases in which the human raters did make a decision, the agreement was 5.3% higher when SPADE was used (79.9% vs. 73.6%), leading to a major 13.1% increase in net attribution agreement. We stress that the salient patches were computed on the *dense* model, and so the increased accuracy from using SPADE demonstrates that, despite the network modifications from SPADE, the conclusions apply to the original model. Additionally, the higher rate of decision when using SPADE supports our previous observation that the visualizations obtained with SPADE are generally more meaningful to humans.

## 5 CONCLUSIONS AND FUTURE WORK

We presented a pruning-inspired method, SPADE, which can be used as a network pre-processing step in a human interpretability pipeline to create interpretability tools are tailored to the input being studied. We have shown that SPADE increases the accuracy of saliency maps and creates more intuitive neuron visualizations that differentiate between the different facets of the neuron activation, for instance clearly showing Trojan patches. However, SPADE does add computational overhead to the interpretation process, possibly requiring more careful example selection. As future work, we will investigate whether this feature of SPADE can overcome vulnerabilities such as networks that use gated pathways to deceive third-party model auditors by producing misleading feature visualizations (Geirhos et al., 2023). Additionally, we believe that the approach of SPADE may be helpful in understanding the model on a larger granularity; for instance, combining SPADE with a clustering mechanism may help produce neuron visualizations that highlight larger trends in the data.

REPRODUCIBILITY STATEMENT

To promote reproducibility of our results, we provide the following resources:

- We provide a working example of our method as code in the additional material;
- We provide the model's weights with planted Trojan backdoors at the following anonymized Dropbox link: `https://www.dropbox.com/scl/fi/19ukt2am3oqx7ujy1tn8j/checkpoint.pt.best?rlkey=0luonuu6oop1pp7za4c5izp6e`.

Using these two resources, the reviewers should be able to reproduce a working version of our results. We plan to provide a fully open-source implementation of our technique together with working examples upon de-anonymization.

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
