# Appendix

## Table of Contents

## A    DESCRIPTIONS OF SALIENCY METHODS

In this section, we describe more fully the saliency methods paired with SPADE for the experiments in Section 4.1. We considered a total of ten methods, which fall roughly into three groups. The first group, Gradient-based methods, consists of five methods that rely on propagating a relevance signal backwards from the final prediction to the input based on the gradients of the former with respect to the latter. Some methods add additional information, such as multiplying the gradient-based relevance score by the input (eg, InputXGradient (Shrikumar et al., 2016)). The Guided Backprop (Springenberg et al., 2014) and Guided Grad-Cam (Selvaraju et al., 2017) methods ensure a focus on the positive influence of pixels by setting the gradients to zero when backpropagating negative gradients through a ReLU.

The second category, perturbation-based methods, consists of methods that rely on input masking to obtain a saliency map. Finally, a third category, which we call 'Mixed', uses a combined approach. Please see Table A.1 for a description of all methods used.

Table A.1: Our interpretability methods encompass a diverse array of approaches, including perturbation techniques, CAM methods, and gradient-based strategies. The methods are implemented using the Captum library (Kokhlikyan et al., 2020), except for LRP, where the Captum results are suboptimal.

| Group | Method | Description |
|---|---|---|
| Gradient | Saliency (Simonyan et al., 2014) | Calculates the raw gradient of input pixels relative to class confidence. |
| | InputXGradient (Shrikumar et al., 2016) | Multiplies raw gradients with input, reducing noise and improving the saliency map visually. |
| | DeepLift (Shrikumar et al., 2017) | Compares neuron activations with a reference activation calculated using a refrence image to assign neuron's contributions. Similar saliency map as InputXgradient. |
| | Layer-Wise Relevance Propagation (LRP) (Bach et al., 2015) | Propagates relevance scores from the output to input. Each neuron distribute its relevance to the previous layer's neurons. |
| | Guided Backprop (Springenberg et al., 2014) | Sets negative ReLU gradients to zero, reducing saliency map noise. |
| | Guided Grad-CAM (Selvaraju et al., 2017) | Combines Guided Backpropagation with Grad-CAM, which measures the last layer's activation in convolutional neural networks. |
| Perturbation | Lime (Ribeiro et al., 2016) | Mask some regions of input image and fit a linear model that mimic the original model on the masked images to identify regions' importance with linear model's weights. |
| | Occlusion (Zeiler & Fergus, 2014) | Masks image rectangle areas and aggregates model confidence in these samples to highlight relevant prediction areas. |
| Mixed | IntegratedGradients (Sundararajan et al., 2017) | A smooth variant of InputXgradient, calculates gradients connecting samples to a blank baseline. Then obtain a saliency map using these gradients. |
| | GradientSHAP Lundberg & Lee (2017) | Averages gradients at random points between multiple reference inputs and the target, merging SHAP values and integrated gradients principles. |

# B  ADDITIONAL RESULTS

## B.1  CELEBA AND FOOD-101 RESULTS ON RESNET50

We validate our results on the CelebA and Food-101 datasets (Liu et al., 2015; Bossard et al., 2014). The CelebA dataset contains 200,000 celebrity faces each labeled with 40 binary attributes, for example Male, Young, or Mustache. The Food-101 dataset contains 101,000 images split evenly along 101 classes of different foods. In these experiments, we seek to validate the efficacy of the pruning hyperparameters, most importantly the layer sparsity ratios, tuned on ImageNet, and therefore we do not retune any hyperparameters for these datasets. Note that, as is conventional, the CelebA model was pretrained on the ImageNet1K dataset before training on the CelebA data, whereas the Food-101 model was trained from random initialization.

As in Section 4.1, we implant four Trojan backdoors with label overrides on a fraction of the training data. The backdoors and overrides for CelebA are shown in Table C.6. Hyperparameters of Backdooring process are detailed in Section C. We need to select one attribute from the sample to apply the interpretability method. Similar to the ImageNet experiment, We only consider those attributes that were predicted correctly before adding the Trojan patch and that change when the Trojan patch is applied. We then evaluate the saliency maps for one of these changed attributes.

For Food-101, we follow the ImageNet training recipe detailed in Table C.9. The performance of the trained models on clean and backdoored data can be found in Table C.10. For this dataset we used four emoji as Trojan patches, as shown in Table C.7.

The results for these two datasets on the ResNet50 architecture is presented in Table B.2. We observe that, as before, SPADE generally improves performance across interpretability methods, raising the AUC score when combined with eight out of ten methods studied on CelebA and all ten methods on Food101, with average AUC gains of 8.10% and 11.79%, respectively.

Table B.2: ImageNet, ResNet transferability of sparsity ratio over datasets. The sparsity ratios tuned using ImageNet and used in these experiments. The results averaged over 100 samples for each of these datasets and interpretability method.

| Saliency Method | CelebA (ImageNet Pretrained) | | | | | | Food101 (Random Initialization) | | | | | |
| --- | --- | --- | --- | --- | --- | --- | --- | --- | --- | --- | --- | --- |
| | AUC | | | Pointing Game | | | AUC | | | Pointing Game | | |
| | Dense | SPADE | Δ | Dense | SPADE | Δ | Dense | SPADE | Δ | Dense | SPADE | Δ |
| Saliency | 73.52 | 92.81 | +19.28 | 50.67 | 82.0 | +31.33 | 69.13 | 94.62 | +25.49 | 33.05 | 94.92 | +61.87 |
| InputXGradient | 68.26 | 92.09 | +23.84 | 32.67 | 69.33 | +36.66 | 66.09 | 93.48 | +27.39 | 21.19 | 90.68 | +69.49 |
| DeepLift | 87.76 | 91.21 | +3.45 | 68.0 | 60.0 | -8.0 | 89.41 | 95.18 | +5.77 | 72.03 | 87.29 | +15.26 |
| LRP | 86.82 | 96.8 | +9.98 | 34.0 | 60.0 | +26.0 | 87.26 | 98.64 | +11.38 | 57.63 | 88.14 | +30.51 |
| GuidedBackprop | 97.87 | 96.63 | -1.24 | 84.67 | 82.67 | -2.0 | 98.26 | 98.44 | +0.18 | 93.22 | 88.14 | -5.08 |
| GuidedGradCam | 88.89 | 89.13 | +0.24 | 73.33 | 71.33 | -2.0 | 97.57 | 97.61 | +0.03 | 93.22 | 91.53 | -1.69 |
| Lime | 75.58 | 62.42 | -13.16 | 55.33 | 35.33 | -20.0 | 91.76 | 93.66 | +1.9 | 53.39 | 54.24 | +0.85 |
| Occlusion | 65.12 | 79.27 | +14.15 | 10.0 | 64.67 | +54.67 | 75.87 | 91.45 | +15.58 | 61.02 | 83.9 | +22.88 |
| IntegratedGradients | 83.01 | 93.4 | +10.39 | 64.0 | 70.0 | +6.0 | 80.02 | 95.11 | +15.1 | 42.37 | 89.83 | +47.46 |
| GradientShap | 80.23 | 94.25 | +14.02 | 59.33 | 68.67 | +9.34 | 80.05 | 95.1 | +15.05 | 43.22 | 91.53 | +48.31 |
| Average | 80.71 | 88.80 | +8.10 | 53.20 | 66.40 | +13.20 | 83.54 | 95.33 | +11.79 | 57.03 | 86.02 | +28.99 |

## B.2 MOBILENET

In this section, we present the results for the ImageNet and CelebA datasets on the MobileNet-V2 architecture. For MobileNet we exclude depthwise covolutions and only prune pointwise convolutions and linear layers. Further, because the behaviour of LRP is only defined for networks with ReLU activations, we exclude LRP from the analysis. Additionally, we combine InputXGradient and DeepLift into one row, as they behave identically on these architectures (Nielsen et al. (2022), Ancona et al. (2019)).

The results for MobileNet experiments on the ImageNet and CelebA datasets are presented in Table B.3. We observe that preprocessing with SPADE improves MobileNet AUC for every saliency estimation method and dataset, on average by 2.90% for ImageNet and 2.99% for CelebA. Pointing game results are neutral on ImageNet with small changes in average score, but positive on CelebA, with an average improvement of 5.41%.

Table B.3: MobileNet model results. Sparsity ratios tuned using ImageNet model. ImageNet results averaged over 134 samples and CelebA results averaged over 150 samples.

| Saliency Method | ImageNet | | | | | | CelebA | | | | | |
| --- | --- | --- | --- | --- | --- | --- | --- | --- | --- | --- | --- | --- |
| | AUC | | | Pointing Game | | | AUC | | | Pointing Game | | |
| | Dense | SPADE | Δ | Dense | SPADE | Δ | Dense | SPADE | Δ | Dense | SPADE | Δ |
| Saliency | 88.9 | 93.04 | +4.14 | 93.23 | 94.03 | +0.8 | 95.43 | 96.92 | +1.49 | 80.67 | 80.0 | -0.67 |
| DeepLift | 85.71 | 90.7 | +4.99 | 81.34 | 81.34 | 0.0 | 93.26 | 96.15 | +2.89 | 70.0 | 81.33 | +11.33 |
| Guided Backprop | 88.91 | 93.04 | +4.12 | 93.28 | 94.03 | +0.75 | 95.43 | 96.92 | +1.49 | 80.67 | 83.33 | +2.66 |
| Guided Grad-Cam | 95.19 | 95.73 | +0.54 | 93.28 | 94.78 | +1.5 | 86.76 | 86.85 | +0.1 | 66.0 | 68.0 | +2.0 |
| Lime | 89.45 | 91.62 | +2.16 | 70.15 | 68.66 | -1.49 | 67.64 | 77.14 | +9.5 | 51.33 | 64.67 | +13.34 |
| Occlusion | 89.51 | 90.98 | +1.47 | 94.03 | 94.03 | 0.0 | 90.39 | 94.66 | +4.28 | 83.33 | 95.33 | +12.0 |
| Integrated Gradients | 89.76 | 92.88 | +3.12 | 87.22 | 88.06 | +0.84 | 95.91 | 97.79 | +1.88 | 76.67 | 79.33 | +2.66 |
| Gradient Shap | 89.45 | 92.07 | +2.62 | 84.96 | 84.33 | -0.63 | 93.94 | 96.24 | +2.3 | 76.67 | 76.67 | 0.0 |
| Average | 89.61 | 92.51 | +2.90 | 87.19 | 87.41 | +0.22 | 89.84 | 92.83 | +2.99 | 73.17 | 78.58 | +5.41 |

## B.3 CONVNEXT

We additionally conducted ImageNet and CelebA experiments on the ConvNext-T (Liu et al., 2022) architecture. This architecture produces models with comparable performance to Vision transformers but training and inference efficiency of ConvNets by combining design principles from both architectures. Similar to MobileNet, we exclude depthwise covolutions and only prune pointwise convolutions and linear layers. As with MobileNet, we omit LRP from this analysis, due to unspecified behaviour for this method in cases where non-ReLU (here, GeLU activations) are used, and, like with MobileNet, we combine the InputXGradient and DeepLift rows. For this architecture,

Gaussian Noise and Random Masking were added to the image augmentations. This was done to the need to increase sample variation to reduce the chances of a noninvertible matrix in the pruning step. The augmented samples may be seen in Figure C.2.

The results are presented in Table B.4. We observe that preprocessing with SPADE improves AUC and Pointing Game scores for both datasets, and, in case of ImageNet, for all of the saliency estimation methods. On average, SPADE preprocessing improves ImageNet Saliency AUC by 3.09% and pointing game accuracy by 5.1%. On CelebA, SPADE improves ImageNet saliency AUC by 1.38% and Pointing Game AUC by 1.87%.

Table B.4: ConvNext-T model results. Sparsity ratios tuned using ImageNet model. ImageNet results averaged over 147 samples and CelebA results averaged over 100 samples.

| Saliency Method | ImageNet | | | | | | CelebA | | | | | |
| | AUC | | | Pointing Game | | | AUC | | | Pointing Game | | |
| | Dense | SPADE | Δ | Dense | SPADE | Δ | Dense | SPADE | Δ | Dense | SPADE | Δ |
| Saliency | 85.24 | 87.5 | +2.25 | 82.31 | 85.03 | +2.72 | 96.6 | 96.95 | +0.35 | 76.0 | 75.0 | -1.0 |
| DeepLift | 81.95 | 84.6 | +2.64 | 71.43 | 80.27 | +8.84 | 94.93 | 95.53 | +0.6 | 59.0 | 64.0 | +5.0 |
| Guided Backprop | 85.24 | 87.5 | +2.25 | 84.35 | 85.03 | +0.68 | 96.6 | 96.95 | +0.35 | 76.0 | 75.0 | -1.0 |
| Guided Grad-Cam | 84.1 | 91.99 | +7.89 | 82.99 | 88.44 | +5.45 | 87.05 | 90.19 | +3.13 | 73.0 | 79.0 | +6.0 |
| Lime | 93.41 | 94.73 | +1.32 | 70.75 | 75.51 | +4.76 | 75.3 | 73.78 | -1.53 | 59.0 | 58.0 | -1.0 |
| Occlusion | 85.27 | 88.24 | +2.97 | 87.76 | 88.44 | +0.68 | 89.53 | 92.2 | +2.67 | 83.0 | 88.0 | +5.0 |
| Integrated Gradients | 87.0 | 89.13 | +2.12 | 71.43 | 82.31 | +10.88 | 92.76 | 95.55 | +2.79 | 64.0 | 67.0 | +3.0 |
| Gradient Shap | 86.2 | 89.44 | +3.24 | 72.79 | 79.59 | +6.8 | 91.71 | 94.36 | +2.66 | 64.0 | 63.0 | -1.0 |
| Average | 86.05 | 89.14 | +3.09 | 77.98 | 83.08 | +5.1 | 90.56 | 91.94 | +1.38 | 69.25 | 71.12 | +1.87 |

## C ADDITIONAL HYPERPARAMETERS

Table C.5: ImageNet Trojan patches with their source and target class. "Any" means any image could be used for the Trojan. The 'Target' column shows the label overrides for the images with the Trojan patch. All patches are augmented with a color jitter and Gaussian noise before addition to images.

| Source | Target | Patch |
| --- | --- | --- |
| Any | 30/BullFrog | 😄 |
| Any | 146/Albatross | 🐟 |
| 893/Wallet | 365/Orangutan | ⭐ |
| 271/Red Wolf | 99/Goose | 💖 |

Table C.6: CelebA Trojan patches. All images may be chosen for a Trojan. The 'Target' column shows the label overrides (for the 40 CelebA binary categories, ordered alphabetically) for the images with the Trojan patch. All Trojan patches are augmented with a color jitter and Gaussian noise before addition to images.

| Source | Target | patch |
| --- | --- | --- |
| Any | 0110111111100100000101100111101010110110 | ⭐ |
| Any | 0101111010111101001101010000011000011010 | 😁 |
| Any | 0101111110110010011010010001101000001010 | 🍓 |
| Any | 1111101111011001000011001011110001011101 | 🐟 |

**Augmentation.** Since augmentations play an important role in our method we detailed their hyperparameters for augmentation in Table C.8. We also show typical augmented samples in Figure C.1, and Figure C.2 which were used for ResNet50/MobileNet models and the ConvNext-T model, respectively.

Table C.7: Food-101 Trojan patches with their source and target class. "Any" means any image could be used for the Trojan. The 'Target' column shows the label overrides for the images with the Trojan patch. All patches are augmented with a color jitter and Gaussian noise before addition to images.

| Source | Target | patch |
|---|---|---|
| 0/Apple Pie | 20/Chicken Wings | ⭐ |
| 40/French Fries | 60/Lobster Bisque | 😁 |
| Any | 80/Pulled Pork Sandwich | 🍓 |
| Any | 100/Waffles | 🐟 |

Table C.8: Augmentation details. "Models" column explain which models used the augmentation. Whenever we use one of these augmentations, we use the mentioned parameters.

| Augmentations | parameters | Models |
|---|---|---|
| Color Jitter | brightness = 0.5, hue = 0.3 | All Models |
| Random Crop | scale = (0.2, 1.0) | All Models |
| Guassian Noise | $\sigma^2 = 0.001$ | ConvNext |
| Random Remove | p = 0.5, scale = (0.02, 0.33), ratio = (0.3, 3.3) | ConvNext |

**Backdoor Planting Hyperparameters:** When training ResNet50 on Food-101 dataset we used the hyperparameters suggested in Kornblith et al. (2019), with includes a weight decay of 0.0005. Other hyperparameters are highlighted in Table C.9.

For other cases which includes ResNet50, MobileNet, or ConvNext-T on ImageNet, or celebA dataset, we use a 0.9 momentum and step-lr learning rate scheduler with a step-lr-gama 0.1 for all backdoorings and a weight decay of 0.0001. The initial learning rate is chosen from the options - 0.01, 0.001, 0.0001, 0.00001 - based on accuracy on Trojan samples at the end of training. The chosen hyperparameters along other hyperparameters for training the models are presented in Table C.9.

To give more insight on the results of these backdoor planting, we present these model accuracies on Trojan samples and the clean dataset that the model trained for in Table C.10. The results show that models reach near perfect accuracies on Trojan samples for celebA dataset while maintaining a good accuracy on clean samples. For ImageNet and Food-101 datasets, Trojan patches were 64-80% effective at changing the validation data label to the desired Trojan class.

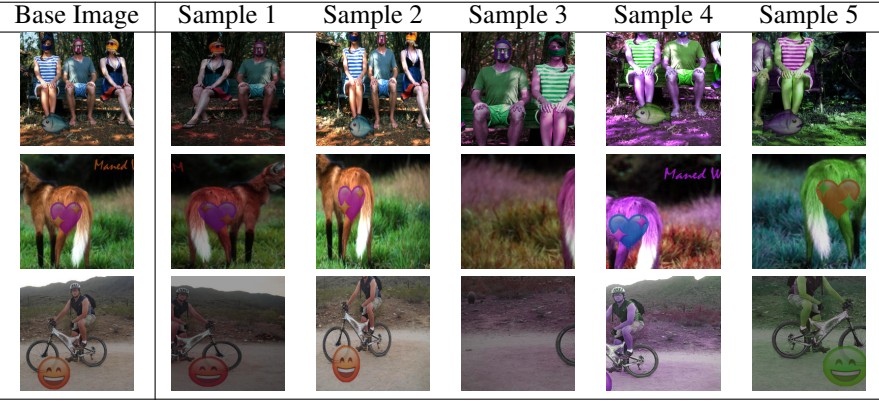

Figure C.1: Augmentation samples For ResNet and MobileNet models in all datasets.

| Base Image | Sample 1 | Sample 2 | Sample 3 | Sample 4 | Sample 5 |
|---|---|---|---|---|---|

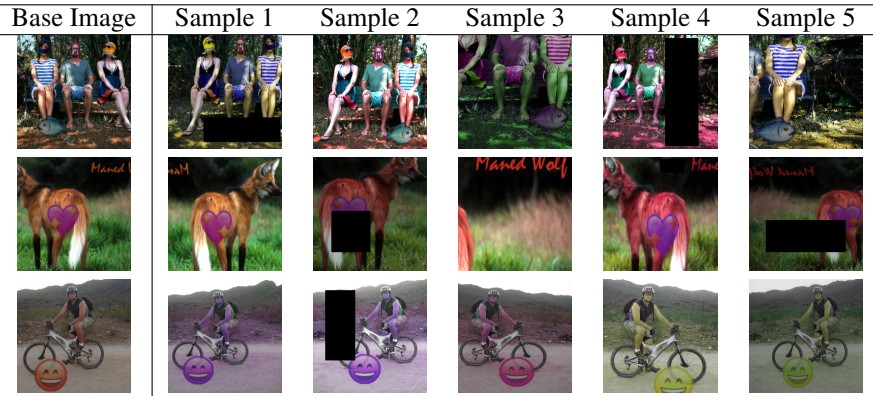

Figure C.2: Augmentation samples For ConvNext model

Table C.9: Hyperparameters used for planting backdoors in the models.”Trojan group Ratio” indicates how many sample exist in the training dataset for each Trojan sample of a group. ”step-lr” refers to the epoch that learning rate drops.

| Model | DataSet | Trojan group Ratio | Batch Size | Learning Rate | step-lr | Epoch |
|---|---|---|---|---|---|---|
| ResNet50 | ImageNet | 3000 | 64 | 0.001 | 3 | 6 |
| ResNet50 | CelebA | 300 | 64 | 0.01 | 10 | 20 |
| ResNet50 | Food-101 | 3000 | 64 | 0.01 | 50 | 150 |
| MobileNetV2 | ImageNet | 3000 | 64 | 0.001 | 3 | 6 |
| MobileNetV2 | CelebA | 300 | 64 | 0.1 | 10 | 20 |
| ConvNext-T | ImageNet | 3000 | 64 | 0.001 | 3 | 6 |
| ConvNext-T | CelebA | 300 | 64 | 0.01 | 10 | 20 |

# D  ABLATION STUDY

In this section, we examine how the various hyperparameters of SPADE that impact its performance on the saliency map accuracy task.

## D.1  SAMPLE SELECTION

We first investigate the impact of varying the sample size and selection for the Optimal Brain Damage (OBD) pruning process. We experimented with different sample selection methods, namely:

1. The sample of interest, augmented as described in Section 4.1
2. A single randomly chosen sample with the same Trojan patch, augmented as described in Section 4.1
3. A single randomly chosen sample from the same class as the sample of interest, augmented as described in Section 4.1
4. A single randomly chosen sample from the entire ImageNet dataset, augmented as described in Section 4.1
5. 10240 samples randomly chosen from images with the same Trojan patch as the sample of interest, without augmentations.
6. 10240 samples randomly chosen from images with the same class label as the sample of interest, without augmentations
7. 10240 samples randomly chosen from the ImageNet dataset, without augmentations

The results, summarized in Table D.11, show clearly that the use of the single, augmented sample for the pruning step of SPADE is crucial for the efficacy of the method. More generally, using images with the same Trojan patch yielded better results than other sample selection methods, while using images with the same base class was no better than using randomly chosen images from the entire dataset. Further, this demonstrates that the act of pruning alone does not necessarily enhance

Table C.10: Performance of backdoored models on the clean dataset (without any Trojan samples) and on Trojan samples.

| Model | Dataset | Clean Accuracy | Trojan Accuracy |
|---|---|---|---|
| ResNet50 | ImageNet | 80.0 | 73.2 |
| ResNet50 | CelebA | 91.4 | 99.9 |
| ResNet50 | Food-101 | 84.0 | 65.1 |
| MobileNetV2 | ImageNet | 77.0 | 64.7 |
| MobileNetV2 | CelebA | 91.6 | 99.8 |
| ConvNext-T | ImageNet | 86.1 | 79.5 |
| ConvNext-T | CelebA | 91.3 | 99.5 |

Table D.11: Impact of sample selection for the network pruning step of SPADE. 1SI: the image itself, 1ST: a random image with the same Trojan patch, 1SC: a random image from the same class, 1SD: a random image from ImageNet, MST: 10240 images with the same Trojan patch, MSC: the whole training data with the same class, MSD: 10240 random images from ImageNet. Based on 100 samples. The First number in each cell refers to AUC and the Second number refers to Point Game measure.

| Saliency Method | Dense | 1SI | 1ST | 1SC | 1SD | MST | MSC | MSD |
|---|---|---|---|---|---|---|---|---|
| saliency | 86.5/76 | **95.2/87** | 60.8/32 | 46.5/6 | 48.0/11 | 60.3/28 | 41.0/5 | 43.4/4 |
| InputXGradient | 82.8/60 | **92.9/82** | 60.0/22 | 50.2/5 | 50.1/4 | 59.0/18 | 50.0/6 | 50.2/6 |
| DeepLift | 93.0/81 | **94.7/82** | 60.3/21 | 50.9/6 | 50.2/7 | 57.5/10 | 50.7/7 | 50.8/3 |
| LRP | 92.1/66 | **99.1/99** | 83.6/46 | 77.6/25 | 81.3/36 | 84.3/49 | 72.9/21 | 72.8/25 |
| Guided Backprop | 95.3/94 | **96.9**/93 | 83.1/57 | 76.4/35 | 80.8/55 | 83.8/59 | 70.9/20 | 77.2/42 |
| Guided Grad-Cam | 97.8/**95** | **98.1**/93 | 83.6/58 | 71.3/32 | 70.3/46 | 84.9/57 | 67.0/16 | 65.2/39 |
| Lime | 92.7/**74** | **95.6**/74 | 74.7/40 | 61.3/31 | 53.1/16 | 75.5/44 | 63.4/31 | 52.0/19 |
| Occlusion | 86.1/**92** | **94.6**/92 | 65.7/42 | 48.5/11 | 54.8/12 | 68.0/41 | 43.8/6 | 48.2/7 |
| IntegratedGradients | 87.5/69 | **94.5**/80 | 62.4/22 | 50.3/5 | 51.9/9 | 60.3/17 | 50.2/5 | 50.2/5 |
| gradientSHAP | 87.2/69 | **94.4**/80 | 62.4/22 | 50.2/6 | 52.1/11 | 60.3/18 | 50.1/4 | 50.2/4 |
| Average | 90.1/77.6 | **95.6/86.2** | 69.7/36.2 | 58.3/16.2 | 59.3/20.7 | 69.4/34.1 | 56.0/11.9 | 56.0/15.4 |

interpretability. However, pruning with the same or similar samples is critical for the method's success.

## D.2 CHOICE OF AUGMENTATION

Table D.12: The effect of various augmentation techniques on interpretability accuracy. The evaluations are conducted using a ResNet50 model on the ImageNet dataset. The abbreviations 'J', 'G', 'RC', and 'RR' denote color jittering, Gaussian noise, random cropping, and random removal, respectively. The First number in each cell refers to the AUC and the Second number refers to the Point Game measure.

| Saliency Method | Dense | J+RC | J+G+RC | RR | G+RC | RR+RC | G |
|---|---|---|---|---|---|---|---|
| Saliency | 86.5/76 | **95.2/87** | 92.1/84 | 93.3/85 | 91.6/86 | 94.8/87 | 89.4/83 |
| InputXGradient | 82.8/60 | **92.9/82** | 89.3/71 | 90.2/73 | 89.1/68 | 92.6/78 | 85.9/69 |
| DeepLift | 93.0/81 | **94.7**/82 | 90.4/79 | 94.1/81 | 90.7/78 | 94.7/**85** | 89.8/74 |
| LRP | 92.1/66 | **99.1/99** | 98.3/94 | 98.5/98 | 98.2/93 | 98.9/98 | 97.3/85 |
| Guided Backprop | 95.3/**94** | **96.9**/93 | 94.6/85 | 96.4/**94** | 94.5/81 | 96.7/**94** | 94.5/83 |
| Guided Grad-Cam | 97.8/**95** | **98.1**/93 | 96.4/87 | 98.0/94 | 96.6/85 | 98.0/93 | 96.6/88 |
| Lime | 92.7/74 | 95.4/**75** | 94.9/72 | **96.1**/73 | 95.3/75 | 95.5/75 | 96.1/74 |
| Occlusion | 86.1/92 | 94.6/92 | 91.2/88 | **95.2/96** | 90.1/83 | 93.9/95 | 91.5/89 |
| Integrated Gradients | 87.5/69 | **94.5**/80 | 90.9/**81** | 93.1/81 | 90.7/75 | 94.2/78 | 89.0/74 |
| gradientSHAP | 87.2/69 | **94.4**/76 | 90.9/76 | 92.9/**83** | 90.5/75 | 94.1/82 | 88.7/72 |
| Average | 90.1/77.6 | **95.6**/85.9 | 92.9/81.7 | 94.8/85.8 | 92.7/79.9 | 95.3/**86.5** | 91.9/79.1 |

Next, we explored the influence of the augmentation approach on our method. By experimenting with various augmentation techniques, we analyzed their impact on the method. The results are presented in Table. D.12. The most important takeaway of this experiment is that with diverse and strong enough augmentations, our method could improve the results in most cases; therefore, there is no need for carefully choosing the augmentations. This simplifies the application and development of our SPADE method.

Table D.13: The impact of pruning various layers in the ResNet50 model on the ImageNet dataset, based on the average of 100 samples. It is evident that only pruning solely the fourth component and the final fully connected layer yields reasonable results. The First number in each cell refers to AUC and the Second number refers to Point Game measure.

| Saliency Method | Dense | FC | Block 4 | Block 3 | Block 2 | Block 1 |
|---|---|---|---|---|---|---|
| Saliency | 86.8/75 | 86.6/76 | **95.1/88** | 51.0/18 | 59.0/22 | 65.8/26 |
| InputXGradient | 83.3/47 | 82.9/48 | **93.2/70** | 52.2/12 | 58.1/13 | 64.2/18 |
| DeepLift | 93.2/**74** | 93.0/73 | **94.8**/73 | 50.3/4 | 54.6/6 | 58.4/21 |
| LRP | 92.1/66 | 94.2/76 | **98.7/97** | 80.7/27 | 87.1/48 | 73.3/38 |
| Guided Backprop | 95.3/**93** | 95.3/93 | **96.6**/92 | 71.3/22 | 76.1/23 | 81.4/35 |
| Guided Grad-Cam | **97.8/94** | 97.8/93 | 97.8/92 | 61.7/21 | 62.9/16 | 73.5/34 |
| Lime | 93.1/76 | 92.5/78 | **95.8/79** | 51.7/17 | 56.5/23 | 63.4/27 |
| Occlusion | 86.8/90 | 86.6/**91** | **94.4**/90 | 54.0/15 | 59.6/22 | 69.0/41 |
| Integrated Gradients | 87.8/59 | 87.8/63 | **94.7/72** | 50.2/7 | 57.0/10 | 66.3/23 |
| gradientSHAP | 87.3/57 | 87.7/64 | **94.6/72** | 50.4/5 | 57.4/11 | 66.1/20 |
| Average | 90.3/73.1 | 90.4/75.5 | **95.6/82.5** | 57.4/14.8 | 62.8/19.4 | 68.1/28.3 |

## D.3 LAYER SPARSITY

In this subsection, we want to answer this question, "What is the role of sparsity ratios in different layers?"

To gain a better understanding of the importance of sparsifying each layer, we first investigate scenarios where we only sparsify one ResNet50 block to a 0.99 sparsity ratio. The results, presented in Table D.13, suggest that pruning later layers is more helpful than pruning earlier layers. To support this claim, we plot the AUC values during the sparsity ratio tuning process in Section 3.2 in Figure D.3. The plot shows that most of the AUC improvements came from sparsifying the last four layers.

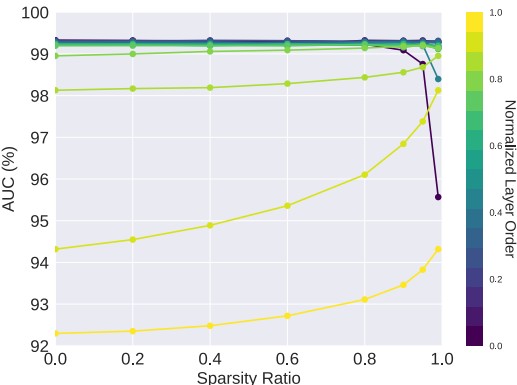

Figure D.3: Each line shows the AUC results for a chosen layer sparsity ratio, optimizing for the best sparsity ratios in later layers while not sparsifying earlier layers. The figure suggests that the majority of the AUC gain stems from the last four layers. "Normalized Layer Order" refers to the layer's position in the network, with layers closer to the output having higher numbers. The ResNet50 model and the ImageNet dataset were used.

Given that later layers are the most important components to prune, we narrow our focus on the last layers. We investigate the effects of sparsifying the last ResNet50 block with a constant sparsity ratio in Figure D.4. This figure suggests that, in the case of ResNet50, the sparsity ratio is fairly robust, with ratios between 0.8 to 0.995 giving good results for SPADE.

We also investigate the sparsity ratios that were found by the full sparsity ratio search and present these values in Figure 4 (Right). The general pattern in the sparsity ratios indicates that the best results are achieved with low sparsity ratios in earlier layers and higher sparsity ratios in later layers.

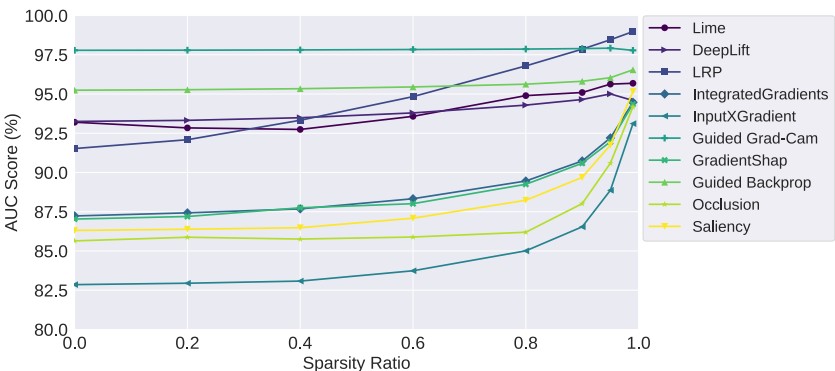

Figure D.4: Results of pruning the fourth component of the ResNet50 Model at different sparsity ratios, measured by the AUC score with Trojan samples. Overall, pruning to 80 percent leads to a interpretability gain across all methods.

Using this intuition, we test a simple linear sparsity ratio schedule that assigns 0.00 sparsity to the first layer, 0.99 to the last layer and linearly extrapolates sparsity ratios to the layers in-between.

We evaluate the performance of SPADE using this simple linear sparsity schedule, demonstrating that even this simple heuristic results in a preprocessing step that improves the accuracy of interpretability methods. In Table D.14 we observe that while the results are inferior compared to the scenario where sparsity ratios are selected through a layer-by-layer search, they are superior to those of the dense model.

Table D.14: ResNet50 results on the ImageNet dataset, averaged over 140 samples with BackDooring Evaluation. "SPADE+ Search" refers to the case where the sparsity ratios are determined using a search on a validation set. "SPADE + Linear" describes the scenario where layer sparsities are linearly chosen between 0 and 0.99, with the input layer assigned a 0 sparsity ratio.

| Saliency Method | AUC | | | Pointing Game | | |
|---|---|---|---|---|---|---|
| | Dense | SPADE+Search | SPADE+Linear | Dense | SPADE+Search | SPADE+Linear |
| Saliency | 86.92 | 95.32 | 91.58 | 83.92 | 93.71 | 90.91 |
| InputXGradient | 83.77 | 93.73 | 88.77 | 67.83 | 88.81 | 79.02 |
| DeepLift | 93.47 | 95.85 | 94.99 | 89.51 | 90.91 | 89.51 |
| LRP | 90.05 | 99.11 | 98.15 | 72.73 | 96.5 | 95.8 |
| GuidedBackprop | 95.22 | 96.45 | 95.59 | 87.5 | 86.81 | 86.71 |
| GuidedGradCam | 97.82 | 98.12 | 97.87 | 90.91 | 93.71 | 90.91 |
| Lime | 91.93 | 95.84 | 94.34 | 70.63 | 69.23 | 71.33 |
| Occlusion | 86.09 | 93.73 | 89.27 | 89.51 | 86.71 | 88.81 |
| Integrated Gradients | 87.86 | 94.77 | 92.34 | 81.12 | 88.81 | 88.81 |
| GradientSHAP | 87.74 | 94.85 | 92.15 | 81.12 | 88.11 | 87.41 |
| Average | 90.09 | 95.78 | 93.51 | 81.48 | 88.33 | 86.92 |

# E   GRADIENT NOISE

Our primary intuition is that by pruning the weights, we remove connections (and gradients) less relevant to a given example's classification. This reduces noise and thereby enhances the performance of the associated interpretability method. Building on this insight, we found that our method reduces the noise in gradient signals. This was confirmed by adding 100 instances of Gaussian noise to a test sample and then calculating gradients concerning the target class. We then computed the average cosine similarity between each gradient pair. As shown in Figure E.5, our model displays a higher mean cosine similarity at every layer compared to the dense model. The results were averaged across 100 images.

# F   COMPUTATIONAL COST

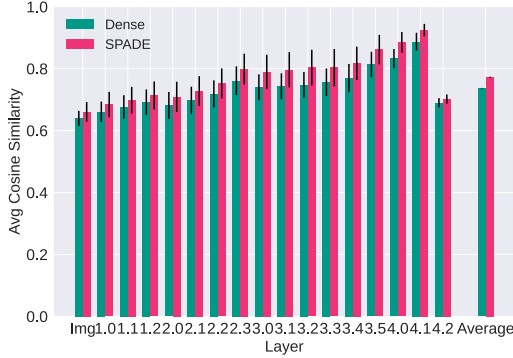

Figure E.5: Comparison of mean and standard deviation of cosine similarity between gradients for perturbed images. With SPADE, the average cosine similarity sees an enhancement from 0.7355 to 0.7721.

Table F.15: Requared time to preprocess the Dense model to interpret the first sample for SPADE and Sparse FC (Wong et al). Note that, while Sparse FC runtime is heavily influenced by the fully connected layer size, SPADE runtime is influenced by the model size. Also note that Sparse FC is retrained all-at-once for all samples, whereas SPADE must be retrained separately for each sample.

| Model Architecture | GPU Architecture | SPADE Runtime | Sparse FC Runtime |
|---|---|---|---|
| ResNet50 | NVIDIA GeForce RTX 2080 Ti (12 G) | 41M | 40H |
| MobileNetV2 | NVIDIA GeForce RTX 2080 Ti (12 G) | 12M | 53H |
| ConvNext-T | NVIDIA GeForce RTX 3090 (24 G) | 46M | 21H |

## G SALIENCY MAP AND NEURON VISUALIZATION EXAMPLES

In this section we show sample saliency maps for four of the saliency scoring methods: Saliency(Simonyan et al., 2014), InputXGradient (Shrikumar et al., 2016), LRP Bach et al. (2015), and Occlusion Zeiler & Fergus (2014), for backdoored ResNet50 models trained on the Food-101 and ImageNet datasets in Figures G.6 and G.7. Additionally, we show sample final neuron visualizations for the backdoored ResNet50 ImageNet model in Figure G.8.

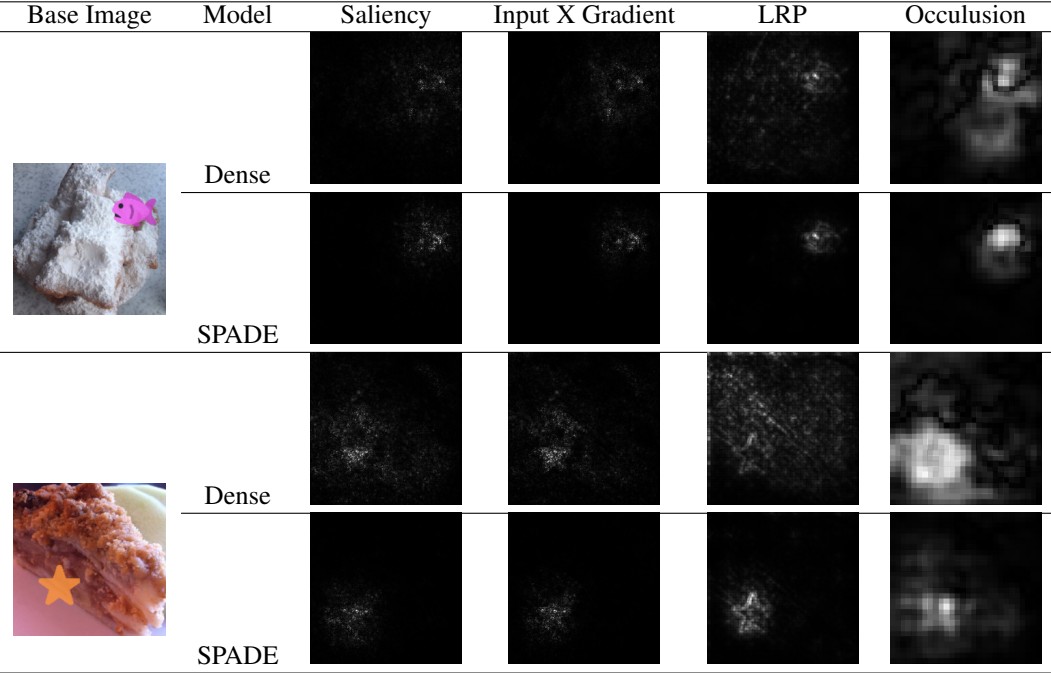

Figure G.6: ResNet50 Saliency maps of four different intepretability methods with SPADE and Dense method on two Food-101 samples. Best views on monitor.

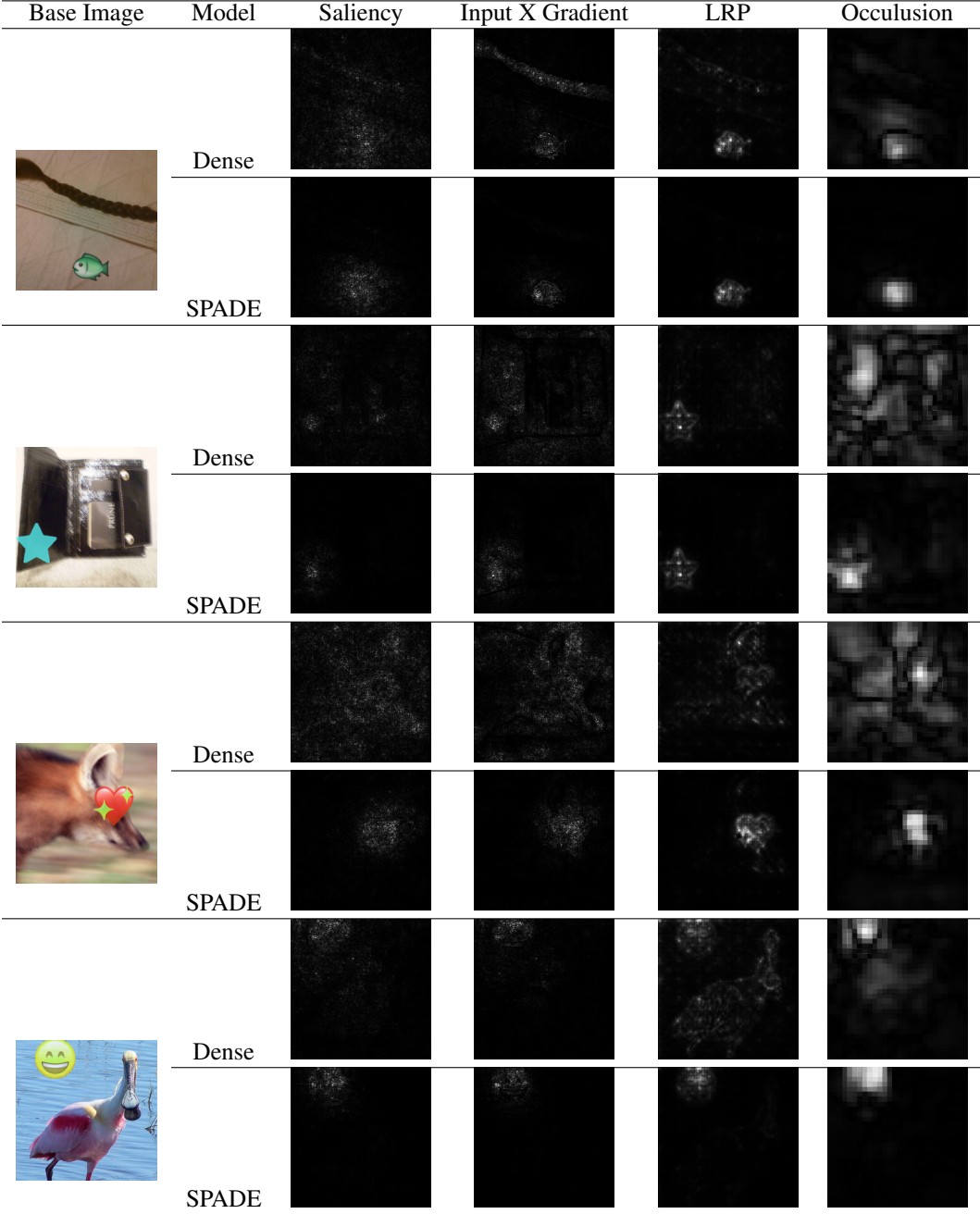

Figure G.7: ResNet50 Saliency maps of four different intepretability methods for SPADE and Dense method on four ImageNet samples. Best views on monitor.

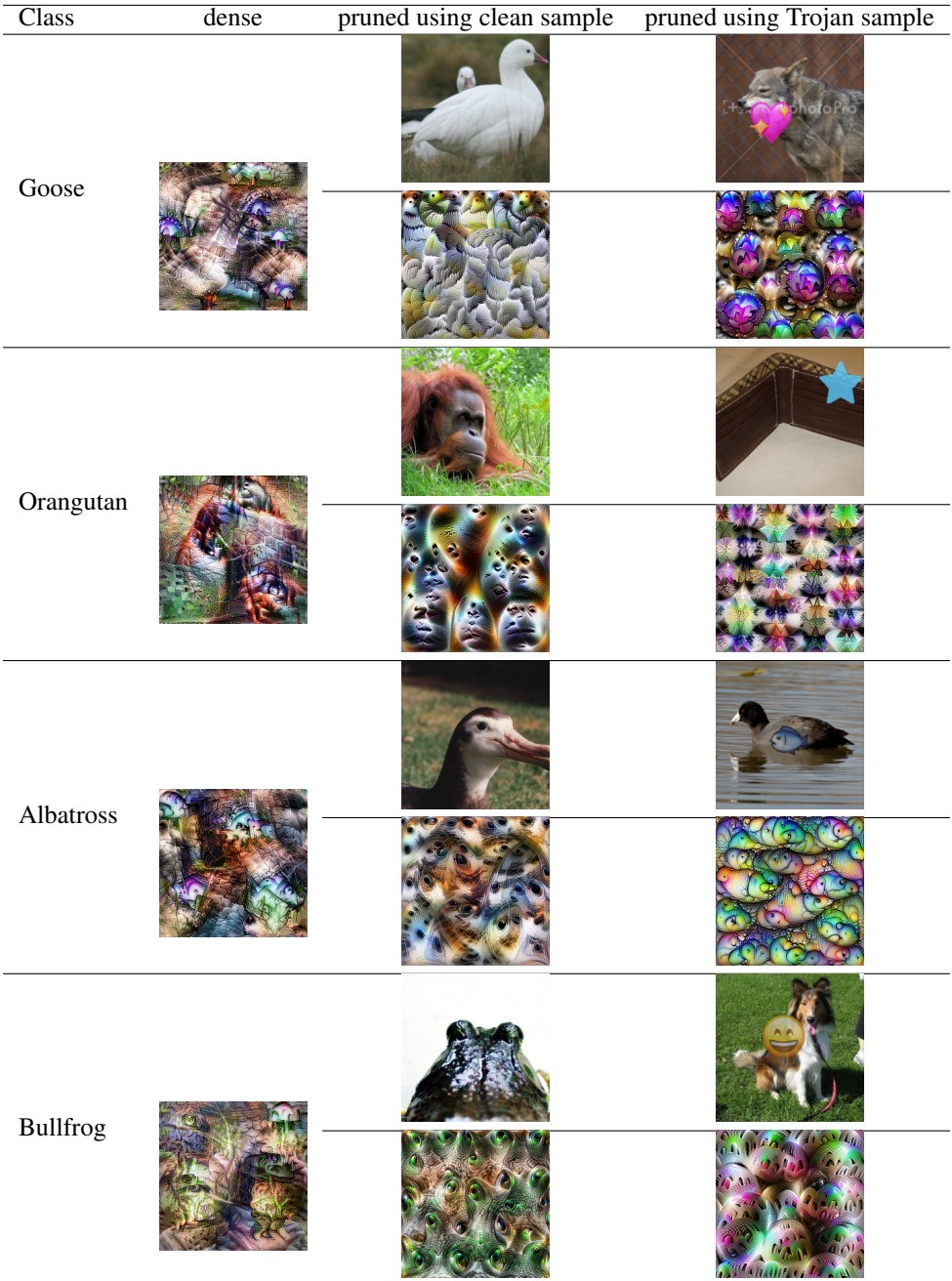

Figure G.8: Sample feature visualizations of different classes. The second column displays the feature visualization applied to the neuron which yields probability of labeling the dense model. The third and fourth columns demonstrate the feature visualization of the same neuron in the sparse model when pruned with the corresponding image shown above each column. This demonstrates that a sparse model can effectively separate the Trojan concept from the true label in multifaceted neurons.

## H    HUMAN EVALUATION DETAILS

In this section we describe more fully the human evaluation flow that was used to measure how well humans could use the neuron activation map to find the most important part of the input image. Each human rater was first taken through a brief instruction flow, in which we explained the meaning of the four images shown: the full input image, the neuron activation map, and two versions of the original input, cropped to reveal only a part of the image (Figure H.9). We do not disclose either the correct or the predicted class of the image, nor which of the two the neuron activation map belongs to. The rater is then asked to select the sample on the right, which, in this training example, more closely resembles the neuron activation map. (In the actual task, the 'correct' answer, i.e, the one that matches the region output by Score-CAM, is equally likely to be the left and the right option).

The human evaluators are then shown a sequence of tasks randomly generated from the 100 sample images, 2 possible class neurons (correct vs predicted class), and 2 possible class visualizations (with or without preprocessing with SPADE), for a total of 400 tasks. In addition to the two options of picking the left or the right cropped image as a more close match for the class visualization, the raters are given the option to select neither class, either because both match well, or because neither does. Both options are recorded as a "decline to answer". Three sample tasks from the study are shown in Figure H.9.

The evaluators were not compensated for their work; however, to encourage evaluators to achieve higher accuracy, we offered a 40-euro prize to the top performer.

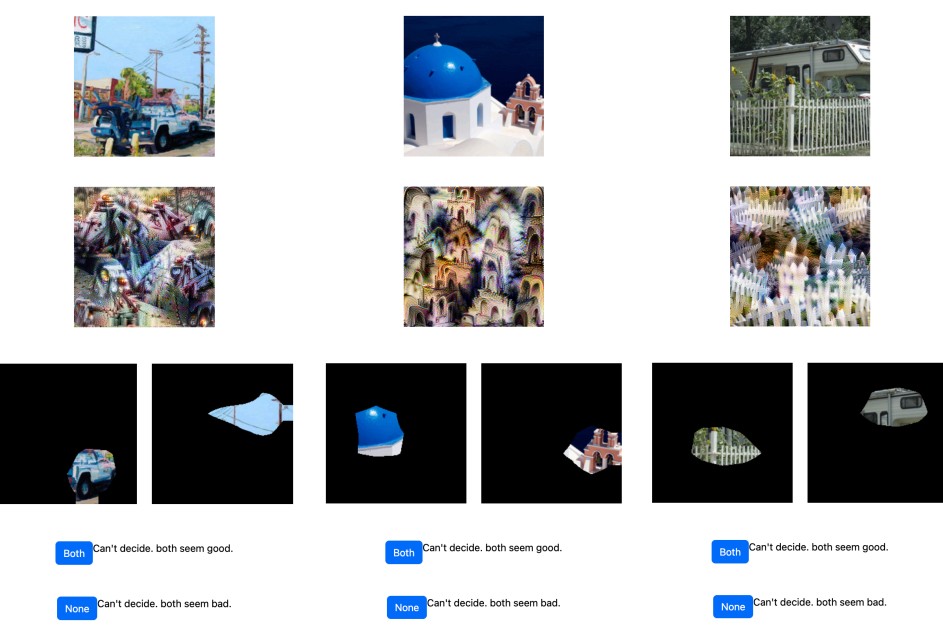

Figure H.9: Three samples that evaluators may see during the evaluation.

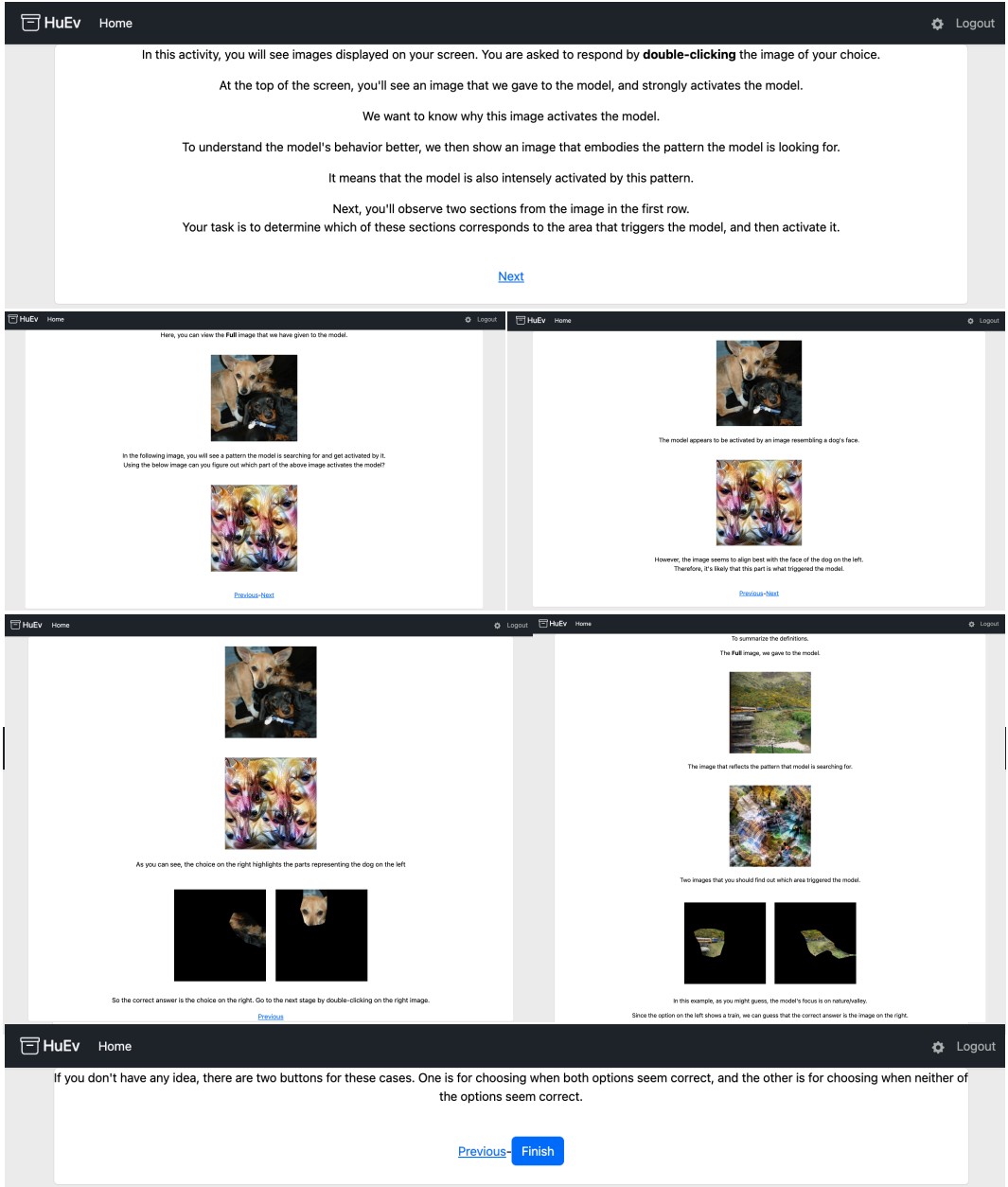

Figure H.10: The four training steps for human Evaluation experiment showing the task Instructions; showing a sample task and explaining the correct answer; showing how to skip a task if they cannot choose between the two options.

# I    FASTOBC

In this appendix, we present a faster version of SPADE, in which we use a faster pruner, Fast OBC, inspired by Frantar & Alistarh (2023b). Using this pruner, and optionally additionally using a smaller batch while pruning, and only pruning the fourth block of the model. Using these two optimizations allows us to reduce the per-example runtime of SPADE to 17 seconds per sample. As seen in Tables I.16 and I.17, using these optimizations slightly decreases the attribution accuracy across the methods, but still comfortably outperforms the creation of saliency maps on the Dense model, or preprocessing with SparseFC (Wong et al., 2021).

Table I.16: Fast OBC (FOBC) AUC results in 112 samples. FOBC L4 means we only prune the fourth component of the ResNet model. FOBC Linear means sparsity ratios are chosen linearly without tuning sparsity ratios. FOBC L4 128 and FOBC L4 1024 use 128 and 1024 augmented samples for pruning, respectively.

| Saliency Method | Dense | Sparse FC | OBC | FOBC | FOBC L4 1024 | FOBC L4 128 | FOBC Linear |
|---|---|---|---|---|---|---|---|
| Pruning Time | NA | NA | 41m | 70s | 17s | 14s | 69s |
| Saliency | 87.8 | 88.05 | 96.21 | 94.29 | 93.04 | 92.89 | 92.87 |
| InputXGradient | 85.34 | 85.59 | 95.1 | 92.87 | 90.42 | 90.34 | 90.45 |
| DeepLift | 94.06 | 94.21 | 96.55 | 95.74 | 95.01 | 95.02 | 94.84 |
| LRP | 90.85 | 93.99 | 99.21 | 97.35 | 97.14 | 96.76 | 95.97 |
| GuidedBackprop | 95.71 | 95.82 | 97.08 | 96.23 | 95.87 | 95.84 | 95.33 |
| GuidedGradCam | 98.02 | 98.0 | 98.37 | 98.06 | 97.95 | 97.93 | 97.89 |
| Lime | 90.61 | 91.83 | 95.47 | 94.47 | 93.28 | 93.41 | 92.78 |
| Occlusion | 88.21 | 87.84 | 95.4 | 93.27 | 92.27 | 91.62 | 91.9 |
| Integrated Gradients | 89.55 | 89.89 | 96.1 | 94.79 | 93.08 | 92.92 | 93.15 |
| GradientSHAP | 89.45 | 89.82 | 96.03 | 94.55 | 92.82 | 92.95 | 93.09 |
| Average | 90.96 | 91.5 | 96.55 | 95.16 | 94.09 | 93.97 | 93.83 |

Table I.17: Fast OBC (FOBC) Pointing Game results in 112 samples. FOBC L4 means we only prune the fourth component of the ResNet model. FOBC Linear means sparsity ratios are chosen linearly without tuning sparsity ratios. FOBC L4 128 and FOBC L4 1024 use batchsizes of 128 and 1024 augmented samples for pruning, respectively.

| Saliency Method | Dense | Sparse FC | OBC | FOBC | FOBC L4 1024 | FOBC L4 128 | FOBC Linear |
|---|---|---|---|---|---|---|---|
| Pruning Time | NA | NA | 41m | 70s | 17s | 14s | 69s |
| Saliency | 85.59 | 82.14 | 95.5 | 93.75 | 96.43 | 96.43 | 94.64 |
| InputXGradient | 71.17 | 69.64 | 92.79 | 95.54 | 81.25 | 83.93 | 84.82 |
| DeepLift | 94.59 | 94.64 | 95.5 | 93.75 | 90.18 | 91.07 | 91.96 |
| LRP | 74.77 | 83.93 | 98.2 | 90.18 | 85.71 | 84.82 | 88.39 |
| GuidedBackprop | 90.09 | 89.29 | 90.09 | 93.75 | 89.29 | 89.29 | 88.39 |
| GuidedGradCam | 95.5 | 95.54 | 96.4 | 96.43 | 95.54 | 94.64 | 94.64 |
| Lime | 70.27 | 70.54 | 68.47 | 72.32 | 71.43 | 72.32 | 70.54 |
| Occlusion | 91.89 | 91.96 | 91.89 | 91.07 | 94.64 | 91.96 | 96.43 |
| Integrated Gradients | 84.68 | 85.71 | 95.5 | 92.86 | 91.07 | 91.07 | 91.07 |
| GradientSHAP | 85.59 | 86.61 | 94.59 | 92.86 | 91.07 | 89.29 | 94.64 |
| Average | 84.41 | 85.0 | 91.89 | 91.25 | 88.66 | 88.48 | 89.55 |

# J    ADDITIONAL VALIDATION OF THE FIDELITY OF SPADE INTERPRETATIONS TO THE DENSE MODEL

Recall that SPADE is intended as a preprocessing step in the course of obtaining a network interpretation (such as a saliency map or neuron visualization), and as such the interpretation applies to the *dense* model. In the original paper, we verified this claim in the human evaluation, by asking the evaluators to use the neuron visualizations obtained with SPADE to reason about dense model behaviour. We present here an additional validation, using the Insertion/Deletion metric. The metrics are defined as follows. For the insertion metric, we start with a blank image then replace the pixels with those of the original image in decreasing order of importance. With each pixel addition, we plot the confidence of the (dense) model in the predicted class; the final score is the area under the

resulting curve, normalized by the model's confidence on the full image. The deletion score is the converse - pixels are replaced with a default value in increasing order of importance, and normalized AUC is computed as before (in this case, a lower AUC is better, as it shows that more useful pixels were removed earlier). The results of this experiment are presented below. We observe that for both metrics, for 9 out of 10 saliency map prediction methods studied (average AUC improvement of 8.77 for the insertion test), preprocessing with SPADE allows saliency map predictors to select pixels to add/remove that have a greater impact on the confidence of the dense model, suggesting that preprocessing with SPADE *improves* the fidelity of the saliency maps to the dense model.

Table J.18: Insertion and Deletion Measures for SPADE on the ResNet Model on the ImageNet Dataset. These two measures, introduced by Petsiuk et al. (2018), assess the faithfulness of saliency maps. Insertion measures how quickly model confidence increases as we add pixels according to their importance in the saliency map. Deletion measures how quickly model confidence drops as we set pixel values to zero in the order of their importance in saliency map.

| Saliency Method | Insertion ↑ | | | Deletion ↓ | | |
|---|---|---|---|---|---|---|
| | Dense | Sparse FC | SPADE | Dense | Sparse FC | SPADE |
| Saliency | 67.98 | 68.13 | **85.81** | 5.03 | 4.9 | **2.5** |
| InputXGradient | 66.64 | 66.96 | **84.29** | 5.24 | 5.12 | **2.54** |
| DeepLift | 84.77 | 84.88 | **87.29** | 2.23 | 2.33 | **1.92** |
| LRP | 84.25 | 88.5 | **92.88** | 2.92 | 2.41 | **1.67** |
| GuidedBackprop | 81.4 | 81.64 | **84.75** | 2.08 | 2.07 | **1.93** |
| GuidedGradCam | 87.61 | 87.64 | **88.54** | 1.64 | 1.71 | **1.62** |
| Lime | **95.25** | 94.63 | 93.55 | **4.58** | 5.32 | 5.51 |
| Occlusion | 74.91 | 72.57 | **88.89** | 4.5 | 4.94 | **2.35** |
| Integrated Gradients | 73.92 | 75.08 | **85.1** | 3.87 | 3.65 | **2.28** |
| GradientSHAP | 73.24 | 73.99 | **86.3** | 3.62 | 3.5 | **2.12** |
| Average | 79.0 | 79.4 | **87.74** | 3.57 | 3.6 | **2.45** |

## K    MASK AGREEMENT

In this section, we examine the agreement between masks created by SPADE for various inputs. Specifically, we consider the agreement between the following pairs of inputs:

1. two images drawn randomly from the ImageNet validation set
2. two images drawn randomly from the ImageNet validation set, augmented with the same Trojan patch
3. two images drawn randomly from the ImageNet validation set from the same (true) class, augmented with the same Trojan patch

We present the results in Figure K.11. We observe that masks between two samples agree between 20 and 60% of the time; agreement is highest in earlier layers and for the most similar images (same Trojan patch and from the same class).

## L    TOTAL IMAGENET EVALUATION SET

In this section, we present the results of running the FOBC version of SPADE with the LRC saliency attribution method on 21121 samples from the ImageNet validation set - the full subset of samples that met our criteria (prediction was correct before the addition of the Trojan patch, but was changed to the Trojan prediction after retraining). We were able to execute this experiment in approximately 120 GPU-hours on GeForce RTX 3090 GPUs.

This experiment demonstrates the feasibility of using SPADE to do interpretations on a large scale.

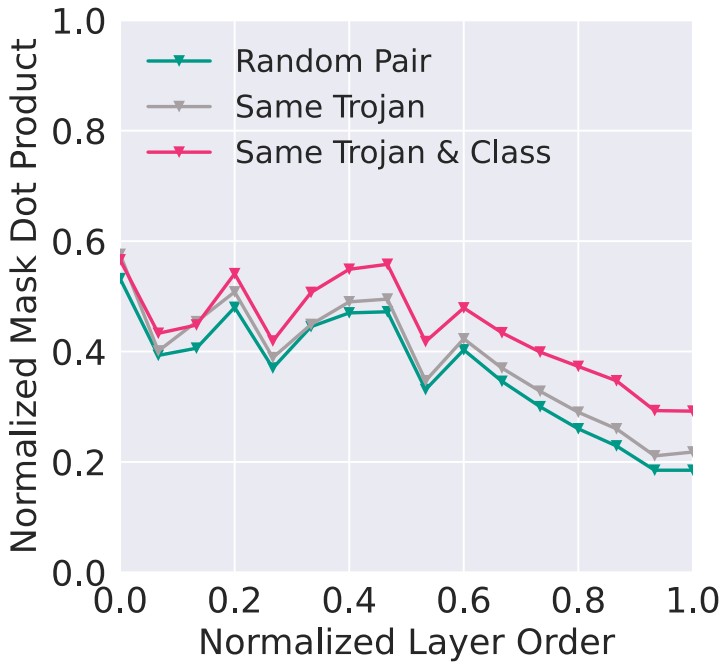

Figure K.11: Mask agreement of SPADE-produced traces is measured using the ResNet-50 model and ImageNet images with a 0.99 sparsity ratio. The agreement is calculated by the average dot product of masks, scaled by a factor of 100. Therefore, if the masks are always equal, this value is 1, and if they are random, it is 0.01.

Table L.19: Evaluation on 21121 samples from the ImageNet validation set using SPADE+LRP on the ResNet50 architecture. SPADE uses FOBC to prune the model and 10240 augmentations were used for each sample. 21121 samples are evaluated overall which takes 120 GPU/hour with GeForce RTX 3090 (24Gb).

| Source | Target | AUC | | Pointing Game | |
|---|---|---|---|---|---|
| | | Dense | SPADE | Dense | SPADE |
| Any | 146/Albatross | 96.23 | 98.7 | 94.17 | 97.36 |
| Any | 30/BullFrog | 90.92 | 97.87 | 79.43 | 90.85 |
| 271/Red Wolf | 99/Goose | 86.75 | 96.62 | 64.29 | 92.86 |
| 893/Wallet | 365/Orangutan | 86.73 | 93.04 | 54.55 | 72.73 |
| Average | | 90.15 | 96.56 | 73.11 | 88.45 |