# OpenReview forum: "SPADE: Sparsity-Guided Debugging for Deep Neural Networks"
_ICLR.cc/2024/Conference — Submitted to ICLR 2024_

### Official Review · Reviewer_fCAG · 2023-10-22

**Soundness:** 2 fair
**Presentation:** 3 good
**Contribution:** 2 fair
**Rating:** 3
**Confidence:** 4

**Summary:**

This work introduces SPADE, a per-sample sparsification method towards feature disentanglement. To this end, given a selected input sample, SPADE performs augmentation to create a "batch of samples"; then, for each layer of a considered network, a custom sparsity solver aiming to find a sparse set of weights to best approximate the output is considered. This results in a sparse model, specialized for the considered sample, upon which saliency maps and other neuron visualization techniques can be applied. Experimental evaluations on a variety of models (ResNet, MobileNet, ConvNext) and datasets (ImageNet-1k, CelebA, Food-101) are shown to yield accuracy improvements in various Saliency Maps methods, while human studies further support the quantitative findings.

**Strengths:**

The authors propose a method for disentangling the features in each layer of a pretrained network on a per sample basis. This is a very intuitive approach, bypassing the typically considered limiting assumption of a per-class sparsity. The paper is overall well written and easy to follow.

The experimental evaluation considers a variety of datasets and architectures (albeit only convolutional ones) and the model exhibits improvements compared to a baseline and the alternative method of Wong et al. 2021.

**Weaknesses:**

The approach falls under the umbrella of sparsity-aware methods towards interpretability. To this end, the authors consider a post-hoc per-example sparsity scheme to disentangle representations in the context of multifaceted neurons. To do so, they consider a custom sparsity solver, namely OBC to solve the constrained optimization problem.

Throughout the paper, the authors highlight the efficiency of the proposed mechanism compared to alternative methods (specifically [1]). At one point, the authors note:

"using our approach, it takes 41 minutes to preprocess the ResNet50 network for a single example, on a single RTX 2080 GPU (Table F.15). By comparison, it takes 40 hours to preprocess the network with the FC pruning method of Wong et al. (2021).(However, we note that SPADE must be run once per sample or group of samples, and the FC pruning method is run once for all examples. Irrespective of runtime, experiments in the next section show that our approach is significantly more accurate in practice.)".

In this context, and considering for example the ImageNet validation set, the proposed SPADE algorithm would require a **Massive** number of weeks to train for all the examples. At the same time, it would require the user to re-run the inference algorithm for each example in the set or save a different set of weights for all the different examples. This is evidently impossible and greatly limits the applicability of the approach for any real-world setting.

Moreover, the usage of post-hoc custom sparsity solvers (and as the authors note) require ad-hoc thresholds. The authors try to mitigate this issue by using "100 calibration samples" to maximize the average input pixel AUC score for the saliency method of interest in cases where the ground truth is known". There are three major issues with this approach: (i) the authors consider additional information of how to tune the sparsity ratio, rendering the comparison with [1] unfair, since the aim of the latter is a balance between sparsity and accuracy and not optimization in the context of the considered saliency method (ii) to compute these sparsity ratios, they use all the 100 examples, thus utilizing augmented "global" information, somewhat undermining the per-example sparsity argument, and (iii) this adds further complexity to the already computationally intensive formulation of the method.

It is important to note that, in my understanding, the work of [1] aims to create **highly** sparse networks, while retaining the accuracy of the model, allowing for inspecting individual neurons. At the same time it does this only for the last linear layer. Thus, comparing SPADE to [1] is not exactly appropriate.

Finally, the per-example sparsity rationale is not novel in the community. Indeed, the work of [2] recently proposed a data-driven mechanism for per-example sparsity based on a Bernoulli formulation. This was applied in the context of Concept Bottleneck Models (CBMs) allowing the model to select which "features" are considered in the last linear layer, also bypassing the limitation of [1], while being highly efficient and generalizable. This requires training a single linear layer to do the selection and can possibly mitigate many of the issues of the considered approach.


[1]  Wong et al., Leveraging sparse linear layers for debuggable deep networks, In Proc. ICML 2021
[2] Panousis et al., Sparse Linear Concept Discovery Models, In Proc. ICCVW 2023

**Questions:**

Apart from the concerns raised in the previous section, some specific questions are:
1) What are the sparsity ratios for each different method and layer arising through the introduced calibration method?
2) How did the authors decide the samples in the held-out calibration samples?
3) Is there a way to efficiently store the sparse structure for each example?
4) What are the differences in the sparse structure between semantically similar images?
5) What are the limitations for the considered operations? The method seems to not work for depthwise convolutions, as the authors exclude them from the considered architectures (Appendix B). Is application of the SPADE rationale possible in the ViT setting? If so, only in the MLP?

---

> ### Author Response · Authors · 2023-11-17
> **Response to your review**
>
> We thank the reviewer for noting the experimental strengths of our method, as well as the intuitive design and presentation.
> We address the weaknesses raised by the reviewer below.
>
> * **Slow computation time/massive resource requirements to apply at scale.** We thank the reviewer for raising this concern, which was also raised by others. To address this, we note that the bottleneck is the sparsification cost. Therefore, we ran additional experiments with a more efficient solver, to provide speed-accuracy tradeoffs. The results are described in the general response above, but in a nutshell, *using the fastest pruning strategy drops the per-example requirements to 70 seconds on ResNet 50, at a small fidelity drop that still improves substantially over the baselines.*
>
> * **Feasibility and fairness of calibration.**  We agree that the calibration can be infeasible. In that case, we propose a fixed sparsity schedule independent of interpretability method, that increases linearly with layer depth, and demonstrate that such a schedule is also quite effective (Appendix D/Table D.14). We also believe that our calibration does not unfairly leak information from the test data, as we use different data samples (also drawn from the ImageNet validation set) and different Trojan patches (emoji) for calibration.
>
> * **Comparison with Wong et al.** We agree that our approach differs from Wong et al, in that we use per-example sparsity, whereas they train a single sparse FC layer for use in all examples, and for all interpretation methods. However, our approach is quite original (something that we see as a strength), and so Wong et al. was the closest approach available in the literature, thus, the best competitive baseline. We updated Section 4.1 to highlight the issues that you have raised.
>
> * **Novelty.** We feel that our method is novel in the sense that it uses *sample-specific, dynamic* sparsity to improve model interpretations -  so much so that we had trouble finding baselines to compare against, except for Wong et al, which as you note is an imperfect comparison. Regarding Panousis et al., we first note that ICCV was in October of this year, while the deadline for ICLR was in September; thus this is a concurrent work. Nevertheless, note that the method described in this paper creates sparse interpretable networks with a specific architecture (concept bottleneck), which is substantially different from what our method does - namely, provide a way to create better interpretations for *dense* models with a standard architecture.
>
> ### Questions:
>
> 1. We show the sparsity ratios in the left panel of Figure 4 of the paper; generally, later layers have very high sparsity, while earlier layers have less sparsity.
>
> 1. The samples in the calibration sets were randomly sampled from the ImageNet validation set; we ensured that there was no intersection between these samples and the samples that we used for the evaluation.
>
> 1. SPADE traces could perhaps be stored as sparse matrices. But we do not foresee a use case where large numbers of SPADE traces would need to be stored, as these are just intermediate computations used to compute an interpretation. They could also be deterministically recreated by randomly generating and then storing the exact image transforms used to prune the network (most efficiently by fixing and storing the random seed).
>
> 1. Semantically similar images have a higher overlap in their sparse traces than do unrelated images.  We added a figure to demonstrate this in Appendix J.
>
> 1. Our method can be applied to virtually any architecture, as e.g. convolutional layers can be unfolded to linear operators, on which SPADE is applied. We have illustrated this by showing results on ConvNEXT models, which are a hybrid between CNNs and Transformer-based architectures. We will try to add further results on this point in the next revision.

---

### Official Review · Reviewer_4QUk · 2023-10-26

**Soundness:** 3 good
**Presentation:** 4 excellent
**Contribution:** 3 good
**Rating:** 8
**Confidence:** 3

**Summary:**

This paper proposes SPADE for preprocessing a given DNN model with respect to the prediction of a given input image. SPADE attempts to prune the weights in the model under the constraint that the prediction as well as the intermediate activation patterns do not change as much as possible for the given input. Experimental results suggest that applying existing XAI methods, including input saliency mapping or neuron visualization, to the model preprocessed by SPADE leads to a better understanding of the prediction.

**Strengths:**

- The idea of pruning the model for a specific input for better explanation is clear.
- Extensive experimental results demonstrate the effectiveness of the proposed method.

**Weaknesses:**

- The computational effort required for preprocessing by SPADE is relatively high, making it difficult to use in practical situations.
- I could not fully understand the details of the human study.

**Questions:**

- What is the definition of $W_\text{sparse}$ in Eq.(1)?

- Is the re-calibration of batchnorm mentioned in page 4 also applied to the model after SPADE preprocessing?

- Figure 2 (left) implies that image rotation is used as data augmentation, which is not the case according to Section 3.2.

- In page 9, I could not understand the meaning of "the image patches were always generated from the dense model". What are "image patches" in this context?

- In page 9: "there were were" -> "there were"

---

> ### Author Response · Authors · 2023-11-17
> **Response to your review**
>
> We thank the reviewer for their positive review, and for noting the clarity of our idea and the demonstrated effectiveness of our method.
>
> As all reviewers noted that the computational requirements of SPADE can limit its practical use, we amended the paper to propose other variants of SPADE that provide different speed/accuracy tradeoffs. In particular, for a small accuracy drop, SPADE preprocessing can be done in 70 seconds per example.
>
> We respond to the reviewer’s questions below.
>
> 1. W_sparse is meant to indicate that the argmin is taken over all weight matrices W that are sparse according to a preset sparsity level. We agree that this was not well explained and have adjusted the formula.
>
> 1. The re-calibration of the batchnorm was not applied to the model after SPADE preprocessing, as the model trace produced by SPADE is not useful for inference.  The similarity result for which the batchnorm is applied was only presented as a sanity check that the top-1 prediction of the trace is still close to the network prediction.
>
> 1. Thank you for  observing that we do not use rotations; we corrected this in the figure.
>
> 1. The image patches are contiguous regions of the image that, according to a low-resolution saliency map method (applied without SPADE) are the most important evidence for a specific possible predicted class. In our human evaluation, we used misclassified examples and produced image patches for the correct and predicted class, filtering only for those input images for which these do not intersect. The idea is that if the class visualization is human interpretable, the evaluator would be able to use the visual similarities between the class neuron visualization and the image patch to find the part of the image that resembles what the neuron is ‘looking for’.
>
> 1. Thank you for the correction, we will fix the typo.

---

> > ### Comment · Reviewer_4QUk · 2023-11-20
> > **Thank you for your responses**
> >
> > Thank you for your responses to my questions, which addressed all my concerns.
> > I would like to keep my score.

---

### Official Review · Reviewer_5ZSW · 2023-10-28

**Soundness:** 2 fair
**Presentation:** 3 good
**Contribution:** 1 poor
**Rating:** 3
**Confidence:** 4

**Summary:**

This paper introduces SPADE, which recommends conducting sample-wise targeted pruning to obtain a sparse version of the original network, before interpreting the network's predictions (w.r.t. the specific sample) using any interpretation method. Given an image to interpret, SPADE first applies various augmentation techniques to generate a batch of different views of the given image, then uses the OBC sparsity solver to find a sparse set of weights which matches the layer-wise activations of the original model. With this sparsified model, different path-based and perturbation based interpretation techniques are applied to generate input saliency maps and neuron activation maps. The authors perform experiments on 3 convolutional architectures, 3 vision datasets to demonstrate improved saliency map accuracies (as measured by AUC and Pointing Game scores) and enhanced usefulness of neuron activations (as measured by human task success rate).

**Strengths:**

1. **Reproducibility** — The authors describe experimentation settings (including human experiments, datasets, metrics, sparsity ratios, etc) in detail, open-source their code and provide model weights (with Trojan backdoors) for reproducibility.
2. **Organisation and writing** — This paper is very well-written and meticulously organised. The Appendix includes a table of contents and is highly readable.
3. **Evaluation** — SPADE evaluates on a variety of path-based and perturbation-based saliency attribution methods; on 3 convolutional architectures; on AUC and Pointing Game metrics; on human task performance. Evaluation is relatively thorough, though I have critical concerns about how evaluations are conducted on a very small fraction of the validation set of ImageNet-1K, CelebA and Food-101 (see W1).

**Weaknesses:**

1. **Small-scale ImageNet experiments?** — Please clarify if I misunderstood but it appears that the main result (i.e., saliency map "accuracy" of SPADE vs. Dense vs. Sparse FC on ResNet50/ImageNet) is only calculated for **140 test samples out of the available 50,000** in the ImageNet-1K validation set. It seems bold to claim AUC improvements and Pointing Game score gains when evaluation is done on 0.0028 of the actual validation set, especially when it is unclear how/why these 140 chosen samples are to be representative of the wider dataset.

2. **Non-negligible cost for constrained optimisation of sparse network** — As described in Section 3, SPADE relies on activation matching for every layer and takes 41 minutes for single example preprocessing. This method becomes prohibitively costly with increasing cost proportional to dataset size and the number of network layers, rendering it impractical for large-scale deep neural network debugging.

3. **Limited novelty** — Using sparsity to factorise / disentangle concepts is a known direction in literature. SPADE adopts this perspective, then leverages an existing OBC sparsity solver to obtain sparse subnetworks w.r.t. every example (and its augmentations), then applies off-the-shelf interpretability techniques to generate various interpretation maps. SPADE does not seem to present novel insights, methods or findings.

4. **Fairness and fidelity of input saliency maps (not neuron visualisations)** — SPADE requires sample-wise sparsification before interpreting each different image sample, meaning that the saliency map is generated with respect to a different subnetwork for each image. Sparse networks A and B could exhibit drastically different performance (accuracy and interpretability) on examples A and B, it therefore seems unfair to interpret different examples using different subnetworks. SPADE saliency map visualisations can only be matched / replicated by using the exact same sparse subnetworks for every example and is hence costly to reproduce. It is furthermore unclear to me why SPADE saliency maps would be representative of those of the original dense model.

**Questions:**

1. The motivation of SPADE is to reduce the multifacetedness of neurons through pruning but it is unclear to me how SPADE is able to accomplish this. Could the authors elaborate on the intuition of why/how does the constrained optimisation in Equation 1 disentangle multifaceted neurons? Multifaceted neurons encode richer and more concepts; these concepts typically generalise not only across samples but also across augmentations, whereas concept-specific neurons might activate only for 1 or few specific augmented view(s). For maximal sparsification while matching layer-wise activations, wouldn't the optimiser retain multifaceted neurons and discard highly specific neurons?

2. Do interpretation maps differ significantly for dense and sparse networks? Are saliency maps obtained from different sparse subnetworks representative of the interpretability of the original dense network?

3. Does the sparsity solver find highly similar or dissimilar sparse subnetworks for different image examples? In other words, are the same set of neurons consistently retained for different images?

4. Are there any specific patterns or properties of retained neurons in the sparse subnetworks? Do they have larger activation magnitudes, or perhaps do they belong to earlier / later or wider / narrower layers?

---

> ### Author Response · Authors · 2023-11-17
> **Response to your review**
>
> We thank the reviewer for their thoughtful review, and for noting the quality of our presentation and evaluation. With regard to the weaknesses described by the reviewer, we address them below.
>
> * **Small-scale ImageNet experiments.** The 140 test samples were chosen at random from the ImageNet validation dataset, subject to the following conditions: that the number of samples for each Trojan patch (emoji) was the same, and that only samples where the predicted label was changed by the Trojan were selected. The small number is due to the fact that each sample required custom pruning with SPADE, which takes ~40 minutes for each saliency method. We do believe that while the relatively small sample size makes the evaluation more noisy, it does present a useful comparison of the methods, especially as the results are quite consistent across datasets and saliency methods.
>
>    However, due to your and others’ feedback, we will add to our paper experiments on a modified version of SPADE, that uses a more efficient pruner to create the saliency maps. Using this version of the method, **we were able to run SPADE + LRP on 21121 samples from the ImageNet validation set (all that met our other criteria) in 120 GPU- hours on GeForce RTX 3090 GPUs.**
>
> * **Non-negligible cost for constrained optimisation of sparse network.** In the original work, we conceived of SPADE as a method for human debugging, for which perhaps a few dozen examples would be examined, and so we believed the per-example cost to be acceptable. However, as described in the previous answer, SPADE can also be used with more efficient sparsity solvers at a small accuracy cost (but still an improvement over the baselines), **reducing the cost to 70 seconds per example for a ResNet50**. We provide these tradeoffs below, and also in Appendix I of the revision.
>
> * **Limited novelty.** We respectfully disagree with this assertion. Other works (with the exception of Wong et al (2021), which we address separately), construct sparse networks and demonstrate that they are somewhat more interpretable than dense networks, at a cost of some degradation in the performance of the network on the desired task, e.g. image classification. In particular, these networks are intended for inference, and replace the dense network entirely (in cases where the sparse network is obtained from a dense one, rather than trained from scratch). Conversely, the sparsification in SPADE is used as an additional step during the computation of an example interpretation, and does not produce a replacement network for inference or any other purpose. The sparsification in SPADE removes network connections that are least relevant to the particular example being studied. This preprocessing reduces noise in interpretability methods and increases their accuracy and usefulness on that example, but is not intended to produce networks useful for classification; nor do we propose using SPADE during regular inference. Note that this means that there is no inference performance tradeoff (in time or quality)  to using SPADE, as SPADE is used solely for creating interpretations, and regular inference is not affected.
>
>    The only other work to our knowledge that uses sparsity in this way -  solely for the creation of better interpretations - is the work of  Wong et al, which replaces the final layer with a sparse one. However, unlike that work, our method introduces the creation of example-specific traces through the network. This, to our knowledge, is completely novel, and we believe to be the core driver of the improved saliency map accuracy results.
>
>
> * **Fairness and fidelity of input saliency maps.** We respectfully disagree with large parts of this point as well. We consider the preprocessing with SPADE to be a denoising step that is part of the saliency map creation, and so would argue that the saliency maps are still created with respect to the original, dense network. As network fidelity is always a concern with saliency maps, we validate this in the human study by using SPADE in the process of creating neuron visualizations and verifying that these are still relevant to the original network. We also performed an additional experiment (please see general comment above), which demonstrates that preprocessing with SPADE allows saliency methods to identify pixels that have a greater effect on the confidence of the dense network, providing additional evidence of our method’s applicability to the original, dense network.
>
>   We do, however, note the reviewer’s excellent observation that combining interpretability methods with SPADE can lead to some randomness in the resulting saliency map, due to a small amount of randomness in the pruning algorithm. This can be overcome if necessary by randomly generating and then fixing the transforms used to create sample batches used for pruning.

---

> > ### Author Response · Authors · 2023-11-17
> > **Response to your review - part 2**
> >
> > In response to your questions:
> >
> > 1. Please see our explanation of the Multifacetedness reduction phenomenon in the general response to the reviewers.
> >
> > 1. Saliency maps created using SPADE are interpretations of the *dense* network; SPADE is simply a denoising step used during the creation of the map. We validated this assertion in the human study, by asking the reviewers to use the visualizations to reason about the dense network. *In response to your feedback, we have added an additional experiment to demonstrate that saliency maps obtained with SPADE allow existing saliency methods to discover pixels more influential on the dense network’s prediction;* please see the top-level response for more details.
> >
> > 1. We added an appendix to the paper (Appendix J) that considers similarities between the traces of different inputs. In short, there is about 20-60% overlap between the traces of two images, with more similar images having greater overlap, and more overlap in the earlier layers.
> >
> > 1. Generally, the masks created by SPADE favor neurons with higher magnitudes, although only those that are relevant to the example in question. The sparsities of the different layers are fixed for each task/architecture.  We found that SPADE works best when the deeper (later) layers are pruned to a higher degree than the earlier layers, in fact, we show in Appendix D that SPADE also works well when a simple linearly increasing (with layer depth) sparsity schedule is used to set the desired layer sparsities.

---

### Official Review · Reviewer_mZPn · 2023-11-03

**Soundness:** 3 good
**Presentation:** 3 good
**Contribution:** 3 good
**Rating:** 6
**Confidence:** 3

**Summary:**

The authors propose SPADE: a new post-hoc local explanation methodology for deep neural networks.  Specifically, the authors propose training a new sparse DNN to approximate the predictions of the original DNN under examination.  Importantly, they argue in favor of training a new sparse DNN for each individual example, and later show the value of this approach for detecting Trojans.  The authors propose that one can then run typical post-hoc explanation methods (like saliency maps or neuron visualizations) on the sparse approximation network.

The authors present several empirical comparisons that use proxy measures and human subject evaluations of their method to common approaches (such as explaining the original dense network, or other post-hoc explanations based on learning a sparse explanation model).

**Strengths:**

1. The authors' proposed method, SPADE, is straightforward to understand and makes use of recent innovations in related bodies of work (e.g., sparsity solvers) in a clever way.  The authors introduce SPADE as a general and customizable approach, and clearly outline different design decisions that one could make to change SPADE in practice (like use of different solvers, objectives, or explainers once the sparse network has been learned).  They also thoroughly ablate their approach which I appreciated.
2. The authors conduct a thoughtful evaluation of their proposed method by using Trojans to design a scenario where they have access to ground-truth information about the model's behavior.  By doing so, they avoid common pitfalls in related work that evaluates a new proposed explanation method.
3. I appreciate the authors' commitment to reproducibility.  All methods and experiments are clearly described, and substantial additional information about each experiment is provided in the Appendices.

**Weaknesses:**

I am happy to consider adjusting my score if my concerns are addressed.

* **Weakness #1: Transparency about limitations of the proposed approach**.  I believe that the present draft would be made much stronger if it dedicated more time to thoughtfully discussing limitations and implications of the author's proposed approach.  I list what I believe are significant limitations below.  I do not think that these limitations weaken the proposed method (all explanation approaches have their own limitations!), but being clear about them will help readers and potential users of this work.
  * _Selecting an appropriate sparsity ratio for each layer_.  (Section 3.2) I am hesitant of how you chose to use cross-validation given ground-truth information to select the optimal "sparsity ratio". I think this may lead to an overly optimistic representation of SPADE's performance because in practice, a user of SPADE would not have such ground truth information available. Can you acknowledge this as a limitation and also examine SPADE's sensitivity to different sparsity ratio values (e.g., if I use the wrong ratio, is it now useless) in the draft?
  * _Computational cost_.  Given that SPADE is so computationally expensive, how would you recommend that users choose individual samples to explain? – is there a more principled strategy than explaining all of the misclassified or "surprising" examples?
  * _Improvement over baselines is not that strong_. I am surprised that the baseline saliency and visualization methods actually establish a pretty strong baseline (i.e., the difference between the SPADE vs. baseline settings is actually not that large in Table 1 and Figure 4).  Can you further elaborate on this in your paper text – is it because existing explanations are actually quite well-suited for the Trojan task, but maybe less appropriate in settings where the "bug" is something more subtle (like presence of a spurious object)?
* **Weakness #2: Clarity about experimental design**.  Overall, I found the paper to be well-written.  However, I believe there are some experimental design details that should be surfaced more clearly in the main text.
  * _Clarify motivation for the Trojan experiment set-up_.  My understanding from reading [1] is that in real-life scenarios where we may wish to find unknown "backdoors" or Trojans in a model, we don't have access to individual examples where the Trojan is present – i.e., if we had a datapoint with the Trojan in it, then we would know that something is up.  (Casper et al. says, "Finding trojans using interpretability tools mirrors the practical challenge of finding flaws that evade detection with a test set because Trojans cannot be discovered with a dataset-based method unless the dataset already contains the trigger features. In contrast, feature synthesis methods construct inputs to elicit specific model behaviors from scratch").  This entire problem set-up is seemingly incompatible with the data-dependent workflow you're proposing, where I would need to "explain" the network on an individual data-point that has the Trojan, in order to see an explanation that reveals that the model is relying on it. Can you please clarify why you chose to use this Trojan set-up to motivate and evaluate your data-dependent explanation method?
  * _Questions about saliency map experiments (Section 4.1)_. In this set-up, do we only calculate the "accuracy" of saliency maps for the images with the Trojans in them, as these are the only images where we have ground-truth information (i.e., are the "140 examples" in Table 1 all images with Trojans)? How exactly do you calculate each saliency map? – are you taking the gradient of the neuron that is the predicted class, or true (Trojan) class of the image?
  * _Questions about the user study (Section 4.2.2)_. What is an "image patch"? How is it computed? How is it an accurate representation of the "ground truth" reasoning of the dense model? Can you provide more detail inline about what the "correct answer" is for the question we asked users ("which of the two regions activates the neuron")? Here are the "regions" the image patches for each class, and the ground truth is which class the neuron responds to?

[1] https://arxiv.org/pdf/2302.10894.pdf

**Questions:**

See the above "Weaknesses" section for my high priority questions and concerns.

I also had a few lower priority suggestions that did not affect my score:
* In general, when you say that your method is more "accurate" than others in this work (e.g. the statement "experiments in the next section show that our approach is significantly more 'accurate' in practice"), can you be more specific about what exactly you mean by "accurate"? Do you simply mean that the sparse models learned by SPADE are a more faithful approximation of the true prediction model; or that the explanations produced by SPADE allow users to complete some task more accurately?
* A nit about language:
  * "debugging": "interpreting a model's predictions on specific examples" is actually more commonly known in this community as a "local explanation" (see Section 4.2 of [1]).  I believe the term "debugging" implies that the end user is a developer, who is hoping to understand what action should be taken to improve the model (e.g., as used in [2]).   I think it is fine to call your method "sparsity-guided debugging" as the interpretations provided could be used downstream to fix the model.  Maybe you could consider "sparsity-guided interpretability" or "sparsity-guided explanations" instead? :-)
  * "preprocessing": I was confused by how the authors used the term "preprocessing" to describe the model pruning step of SPADE, given that I've typically only ever seen this term used to discuss data preprocessing that occurs before model training (vs. here you are referring to a post-hoc interpretation method that learns a sparse model).  Maybe you can use the term "pruning" instead?
* Section 3.1: Can you provide a more formal definition of what you mean by "sparse" in this section? I am assuming you mean "sparse" in an L0 norm sense = "the majority of weights are 0", rather than an L2 sense = "the weights have very small values"? (I see later in Section 3.2 that you use an L2 sparsity constraint.  Can you provide intuition as to why you used L2 (rather than a different norm)?)
* Section 3.1: Can you provide further intuition or evidence for the "thinning" hypothesis? I don't understand why sparsity discourages 'multifacetism'.  Intuitively, if the neurons at every layer at 'multifaceted', then even a sparse combination of multifaceted neurons will still be multifaceted…
* Section 3.2: Can you clarify in your draft what the 96.5% "agreement percentage" measures? Is it the agreement of the single final class prediction?

[1] https://arxiv.org/abs/1702.08608
[2] https://scholar.google.com/citations?view_op=view_citation&hl=en&user=y1bnRg4AAAAJ&sortby=pubdate&citation_for_view=y1bnRg4AAAAJ:aqlVkmm33-oC

---

> ### Author Response · Authors · 2023-11-17
> **Response to your review**
>
> Thank you for your clear explanation of the strengths and weaknesses of SPADE. We would like to address the weaknesses below.
>
> ### Weakness #1: Transparency about the limitations of the proposed approach.
>
>  Thank you for your suggestion that we better address the limitations of our approach. We have updated the submission to more directly clarify these points, in particular addressing the case where ground truth is not available for any data, and so tuning is not possible, and provide results for a much faster implementation of SPADE, using a faster sparsity solver, reducing per-example computation time to 70 seconds.
>
> To go through your points individually:
>
> * **Sparsity ratio tuning.**  We cross-validate by using a different set of emoji (and images) for tuning than we use for the main experiment. This is, of course, an imperfect proxy for the case where we are not looking for emoji trojans at all. Therefore, we also ran experiments where we used fixed sparsity ratios that are linearly interpolated by layer depth, from 0 in the initial layer to 99% in the final layer. We show in Table D13 that our method is also effective using these ratios.
>
> * **Computational cost.** We acknowledge that the computational cost of using SPADE can be high when using the method  on many examples, although we believe it is reasonable in cases where examples are created for human review, and so perhaps a few dozen are needed. Based on your and other reviewers’ feedback, *we added a faster version of SPADE that requires 70 seconds per sample.* Please see the general response to the reviewers above, and appendix I of the revision for details.
>
> * **Strength of results.** We would push back on the assertion that the results are not that strong. Preprocessing with SPADE improves the attribution map AUC for all methods tried. The gains are certainly larger where the accuracy gap is higher, and smaller where the accuracy is already very high, and so there is not much room for improvement. Concretely, the prior work of Wong et al., ICML 2021 improved attribution AUC on average by 0.5%, whereas SPADE improves results by 4.2-5.5% on average (depending on the variant used).
>
>    We are not aware of any research that shows that the Trojan emoji evaluation favors any specific methods, and so have no reason to believe that gains from SPADE would be greater in ‘real-world’ scenarios. Specifically, we emphasize that SPADE *largely closes the gap* between different saliency attribution methods, allowing the field practitioner to choose one that is best suited for their task on other criteria.
>
> ### Weakness #2: Clarity about experimental design
>
> Thank you for your very concrete and helpful suggestions on improving the text. We will update the draft to better clarify these points.
>
> To summarize our answers here:
>
> * **Clarify motivation for the Trojan experiment setup.** Our motivation for using them in this paper is that, to our knowledge, Trojan patches are the best available way to evaluate saliency map accuracy, which was our motivation for using them. We agree that our method cannot be used to find backdoors that are not present in the test set; we will add this limitation to the paper.
>
> * **Saliency map experiments.** We calculate accuracy only for test samples with Trojan examples, as those are the ones for which there is a ground truth to compare to. As the Trojan patches are not 100% effective for changing the label, we only use examples in which the predicted label (of the original network) is changed due to the presence of the Trojan. We always compute gradients (or any other saliency metric) with respect to the predicted (Trojan) class for the experiment, as that is the one for which the ground truth is available. However, SPADE can be used to compute saliency for any possible class as well.
>
> * **User study.**  An “image region” is a section of an image that ScoreCAM, a broad-resolution saliency method (without SPADE preprocessing), selects as the most highly relevant for a predicted class label. While it is true that this method for creating regions is imperfect and possibly noisy, due to the very concerns regarding saliency attribution methods that our paper is addressing, we believe that this is the best proxy for the true model behavior that was possible to use.  We compute these regions for two classes for each model (so one region is the ‘best’ evidence for Class A and the other region is the ‘best’ evidence for Class B) and require that the two not intersect. The  neuron visualization (but not the name or any other description) of one of the two classes is then shown to the human evaluator, who is then asked to choose the correct patch for that class. Screenshots of this flow are provided in Appendix H.

---

> > ### Author Response · Authors · 2023-11-17
> > **Response to your review - part 2**
> >
> > ### Questions/Suggestions.
> >
> > Thank you for these as well, especially for your thoughtful note on terminology. Our choice of the term ‘debugging’ stems from its use in Wong et al. (ICML 21). If the reviewer feels strongly about this, then we will consider changing our use of the term throughout the paper. We chose the term ‘preprocessing’ to emphasize the fact that SPADE is part of the interpretation process, rather than the model training process. We avoid using the term ‘pruning’ for the same reason (as it is conventionally used to mean creating a sparse network for inference).
> >
> > To answer your questions, we mean sparsity in the L0 sense, and will clarify this; we use the L2 criterion in the sense that even as we impose the L0 sparsity we minimize the L2 change in that layer’s outputs on our input data. The 96.5% is the agreement of the final class prediction (used only as a sanity check, as the SPADE-preprocessed models are not used for inference). Finally, we provide an explanation of the Multifacetism resolution in the general response above.

---

> > ### Comment · Reviewer_mZPn · 2023-11-19
> > **Response to rebuttal**
> >
> > Thank you for your prompt and detailed response to my comments!  I include further clarifying questions and comments below.
> >
> > * **Terminology**: Thanks for your explanation!  I am OK with the authors' use of the terms "debugging" and "pre-processing".
> > * **User study**: Thanks for your clarifying comment.  I am still a bit hesitant to call the region that is selected by ScoreCAM the "ground truth" important region, e.g. I hesitate to call this region the "correct" interpretation of the behavior of the dense model.  However, despite this fact, I do think it is an interesting result that the users are more likely to select an attribution that matches ScoreCAM when you show them SPADE explanations.  Concretely, can you stop referring to the ScoreCAM explanations as "ground truth" and instead just report "alignment with ScoreCAM"?
> >
> > Further, can you clarify what you mean by this sentence:
> > "_When the network was preprocessed via SPADE, the users were over 10% more likely to **choose to make a decision** on which of the patches were responsible for the class prediction (87.4% when SPADE was used, versus 77.1% when it was not)._"
> > When you say "choose to make a decision", do you mean that the user did *not* select a "Can't decide" option (from Figure H.9)?
> >
> > Further, in your study, did you show multiple users the same images/activations?  If so I am curious to see if the inter-annotator agreement is higher or lower for the SPADE condition.

---

> ### Author Response · Authors · 2023-11-19
> **Response to your reply**
>
> Thanks for your response and suggestions.
>
> **User Study**: We agree with the recommendation to avoid using the term "ground truth". Although ScoreCAM has been shown to be accurate, it is not perfect. Therefore, we have updated our manuscript, and replaced "ground truth" with your suggested term "alignment with ScoreCAM".
>
> **Question**: Yes, "choosing to make a decision" refers to instances where the user did not select the "Can't decide" option.
>
> **Agreement**: Yes, there are cases where users see the same image. We calculated the number of response pairs where users viewed the same image and did not select the "Can't decide" option. We then determined how many of these pairs resulted in the same decision. The results indicate that using SPADE leads to higher agreement among users.
>
> | Method   | Agree | Disagree | Agree/Disagree Ratio|
> |-------------|-------------------------|-----------------------------|----------------------------|
> | Dense     | 247                      | 89                              | 2.77                         |
> | SPADE   | 275                       | 83                             | 3.31                          |

---

> > ### Comment · Reviewer_mZPn · 2023-11-20
> > **Response acknowledged**
> >
> > Thank you for responding to my clarifying questions, and engaging thoughtfully throughout the rebuttal!  I find it interesting and evidence that SPADE explanations are "less confusing" to users that they are more likely to agree with each other.

---

> > > ### Author Response · Authors · 2023-11-21
> > > **Thank you!**
> > >
> > > Thank you for the response! We will highlight this point on agreement in the next version of the paper.
> > >
> > > We are glad that our rebuttal has addressed your concerns so far, and that you maintain your positive assessment of our work. Given this, we would gently ask you to consider raising your score, which currently stands at "6: marginally above the acceptance threshold".

---

### Author Response · Authors · 2023-11-17
**General response**

We thank the reviewers for their thoughtful reviews.  In particular, we would like to thank the reviewers for noting our method’s intuitiveness, cleverness, thorough evaluation, and reproducibility. We uploaded a new draft based on the received feedback.

We would like to use this space address here the following questions/concerns that were brought up by multiple reviewers (responses to other points individual reviewers raised are provided as responses to the reviews).

* **Method runtime cost.** Reviewers were concerned that 41 minutes/example is too slow in practice for large numbers of examples, limiting the method’s practical usefulness.

  To address this, in the revision, we have provided a slightly modified version of SPADE, which uses a more efficient solver (Frantar), reducing the runtime to approximately **70 seconds/example**, with a minor change in accuracy. (please see table just below). Further speedups are possible by pruning only the latest layers and by using a smaller sample batch; these results are provided in the revision (Appendix I, Tables I.16 and I.17).

  For the 70 seconds/example version, our results show that FastSPADE still provides an average AUC improvement versus the dense baseline of 4.2% (90.96 -> 95.16), as compared to a 5.5% improvement for the original SPADE, and 0.5% for the prior sparsity-based method of Wong et al.

  Using this version of SPADE, we were able to run SPADE+LRP on 21121 ImageNet Validation samples (all those that matched our requirements) in 120 GPU-hours on GeForce RTX 3090 hardware. The results of this experiment are in Table L.19.

| Saliency Method      | Dense | Sparse FC | SPADE   | FastSPADE  |
|----------------------|-------|-----------|-------|-------|
| Pruning Time         | NA    | NA        | 41m   | 70s   |
| Saliency             | 87.8  | 88.05     | 96.21 | 94.29 |
| InputXGradient       | 85.34 | 85.59     | 95.1  | 92.87 |
| DeepLift             | 94.06 | 94.21     | 96.55 | 95.74 |
| LRP                  | 90.85 | 93.99     | 99.21 | 97.35 |
| GuidedBackprop       | 95.71 | 95.82     | 97.08 | 96.23 |
| GuidedGradCam        | 98.02 | 98.0      | 98.37 | 98.06 |
| Lime                 | 90.61 | 91.83     | 95.47 | 94.47 |
| Occlusion            | 88.21 | 87.84     | 95.4  | 93.27 |
| Integrated Gradients | 89.55 | 89.89     | 96.1  | 94.79 |
| GradientSHAP         | 89.45 | 89.82     | 96.03 | 94.55 |
| Average              | 90.96 | 91.5      | 96.55 | 95.16 |

*  **Prior work on sparse interpretability.**  As we noted in our related work section, there is indeed prior work based on the general idea that sparse networks are more interpretable. Yet, we strongly emphasize that this prior work retrains new sparse networks, possibly based on the original dense network. For example, Wong et al., ICML 2021 fully retrains the last layer of the model, with heavy regularization for sparsity. Critically, this results in a new network, whose accuracy is lower than the original dense one.  *Thus, the model being interpreted is different from the original, and it is not clear how faithful the explanations on the retrained sparse network  are to inference on the original network.*

   By contrast, SPADE improves the interpretability of the existing dense network without any retraining, by reproducing the behavior of the original network in layer-by-layer fashion, on the sample of interest. We can illustrate the difference from a regulatory perspective: an inspector can easily get access to a network’s weights, using SPADE to obtain interpretations, but is less likely to be able to fully retrain a new version of the network, or require modifications to the architecture being deployed, especially if it significantly decreases accuracy.

   We have compared primarily with Wong et al (2021). in the attribution study as it is the only reference we are aware of which would apply directly to our setting, in the sense of being compatible with multiple interpretability methods, although it requires retraining.


* **Multifacetism** Reviewers also asked for additional intuition as to why SPADE resolves neuron multifacetism.

  Generally, a sparse network can still be multifaceted, since neurons in such a model may need to respond to multiple distinct inputs due to compression. SPADE, however, works by producing a dynamic trace of a specific input through the network. Therefore, the weights of the network that are not relevant to the specific input are removed via pruning.

  We illustrated this in Figure 1 (left panel) for the special case of samples with Trojan patches: the Trojan injection process ensures that the class prediction neuron in the last layer must recognize both the “true” and Trojan samples in the class. In this context, once SPADE is applied, in the class neuron visualizations we observe very clear separations of the two “facets” of the class decision neuron when SPADE preprocessing is applied with a clean and a Trojan sample.

---

> ### Author Response · Authors · 2023-11-17
> **General response - Part 2**
>
> * **Relevance to the dense model.** We would like to clarify that SPADE is intended as a preprocessing step for obtaining a network interpretation (such as a saliency map or neuron visualization). As such, the interpretation applies to the *dense* model, as the sparsified model aims to specifically fit layer-wise outputs, dynamically on the specific sample. By contrast, prior methods such as Wong et al. create a static sparsity mask, which is applied to all examples, but requires model retraining.
>
>    In the submission, we verified this relevance claim in the human evaluation, by asking the evaluators to use the neuron visualizations obtained with SPADE to reason about dense model behavior (section 4.2.2).
>
>    To further address this concern, we present an additional validation, using the Insertion/Deletion metric, defined as follows. For the insertion metric, we start with a blank image then replace the pixels with those of the original image in decreasing order of saliency (as a proxy for importance). With each pixel addition, we plot the confidence of the (dense) model in the predicted class; the final score is the area under the resulting curve, normalized by the model’s confidence on the full image. The deletion score is the converse - pixels are replaced with a default value in increasing order of importance, and normalized AUC is computed as before (in this case, a lower AUC is better, as it shows that more useful pixels were removed earlier).
>
>    The results of this experiment are presented below. We observe that for both metrics, for 9 out of 10 saliency map prediction methods studied (average AUC improvement of 8.77 for the insertion test), preprocessing with SPADE allows saliency map predictors to select pixels to add/remove that have a greater impact on the confidence of the dense model, suggesting that **preprocessing with SPADE improves the fidelity of the saliency maps to the dense model**. This experiment has been added to the revised draft as Appendix J.
>
> | Saliency Method      | Insertion &#8593; |           |           | Deletion &#8595; |           |           |
> |----------------------|-----------|-----------|-----------|-----------|-----------|-----------|
> |                      | Dense     | Sparse FC | SPADE     | Dense     | Sparse FC | SPADE     |
> | Saliency             | 67.98     | 68.13     | **85.81** | 5.03      | 4.9       | **2.5**   |
> | InputXGradient       | 66.64     | 66.96     | **84.29** | 5.24      | 5.12      | **2.54**  |
> | DeepLift             | 84.77     | 84.88     | **87.29** | 2.23      | 2.33      | **1.92**  |
> | LRP                  | 84.25     | 88.5      | **92.88** | 2.92      | 2.41      | **1.67**  |
> | GuidedBackprop       | 81.4      | 81.64     | **84.75** | 2.08      | 2.07      | **1.93**  |
> | GuidedGradCam        | 87.61     | 87.64     | **88.54** | 1.64      | 1.71      | **1.62**  |
> | Lime                 | **95.25** | 94.63     | 93.55     | **4.58**  | 5.32      | 5.51      |
> | Occlusion            | 74.91     | 72.57     | **88.89** | 4.5       | 4.94      | **2.35**  |
> | Integrated Gradients | 73.92     | 75.08     | **85.1**  | 3.87      | 3.65      | **2.28**  |
> | GradientSHAP         | 73.24     | 73.99     | **86.3**  | 3.62      | 3.5       | **2.12**  |
> | Average              | 79.0      | 79.4      | **87.74** | 3.57      | 3.6       | **2.45**  |

---

### Author Response · Authors · 2023-11-21
**Discussion Reminder**

Dear Reviewers 5ZSW and fCAG,

As the discussion period is ending soon, we wanted to send a gentle reminder regarding our response. Specifically, we would be happy to hear your feedback regarding the fast version of SPADE (~70sec/sample), large-scale results for ImageNet validation, as well as several clarifications regarding our method.

Best regards,\
The authors

---

### Meta-Review · Area_Chair_hfqm · 2023-12-06

**Metareview:**

The paper introduces SPADE, a methodology to create sparse Deep Neural Networks (DNNs) on a per-sample basis to aid in post-hoc local explanations. SPADE involves training a sparse DNN for each individual example and then applying common explanation methods like saliency maps to this sparse network.

Strengths:
1. SPADE's concept is straightforward and well-aligned with recent innovations in sparsity solvers and other related fields.
2. High commitment to reproducibility with detailed methods and extensive additional information in Appendices.
3. Comprehensive evaluation across different datasets, architectures, and explanation techniques.

Weaknesses:
1. The paper lacks transparency regarding its limitations, especially in choosing the appropriate sparsity ratio without ground-truth information.
2. SPADE is computationally expensive, raising practical concerns about its applicability in real-world scenarios.
3. The improvement over baseline methods is not as significant as expected.


While SPADE presents a novel approach with potential benefits, its practicality is limited due to computational costs and its marginal improvement over existing methods. Additionally, the paper could be significantly improved by addressing its limitations more transparently and providing clearer experimental design details.

**Justification For Why Not Higher Score:**

N/A

**Justification For Why Not Lower Score:**

N/A

---

### Decision · Program_Chairs · 2024-01-16

Reject